# AssistanceZero: Scalably Solving Assistance Games

**Cassidy Laidlaw** [1]  **Eli Bronstein** [1]  **Timothy Guo** [1]  **Dylan Feng** [1]  **Lukas Berglund** [1]  **Justin Svegliato** [1]
**Stuart Russell** [1]  **Anca Dragan** [1]

## Abstract

Assistance games are a promising alternative to reinforcement learning from human feedback (RLHF) for training AI assistants. Assistance games resolve key drawbacks of RLHF, such as incentives for deceptive behavior, by explicitly modeling the interaction between assistant and user as a two-player game where the assistant cannot observe their shared goal. Despite their potential, assistance games have only been explored in simple settings. Scaling them to more complex environments is difficult because it requires both solving intractable decision-making problems under uncertainty and accurately modeling human users' behavior. We present the first scalable approach to solving assistance games and apply it to a new, challenging Minecraft-based assistance game with over $10^{400}$ possible goals. Our approach, AssistanceZero, extends AlphaZero with a neural network that predicts human actions and rewards, enabling it to plan under uncertainty. We show that AssistanceZero outperforms model-free RL algorithms and imitation learning in the Minecraft-based assistance game. In a human study, our AssistanceZero-trained assistant significantly reduces the number of actions participants take to complete building tasks in Minecraft. Our results suggest that assistance games are a tractable framework for training effective AI assistants in complex environments. Our code and models are available at https://github.com/cassidylaidlaw/minecraft-building-assistance-game.

[1]University of California, Berkeley, CA, USA. Correspondence to: Cassidy Laidlaw <cassidy_laidlaw@cs.berkeley.edu>.

*Proceedings of the 42nd International Conference on Machine Learning*, Vancouver, Canada. PMLR 267, 2025. Copyright 2025 by the author(s).

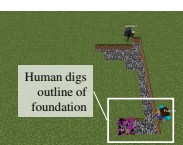 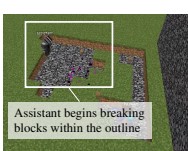 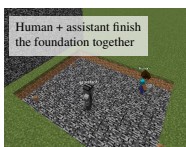

**Digging a foundation:** the assistant watches the human outline the house's foundation and then digs it out.

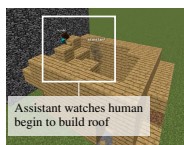 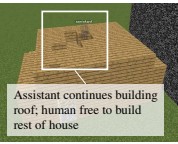 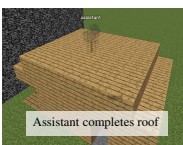

**Building a roof:** the assistant infers the structure of the roof from human actions and completes it while the human builds another part of the house.

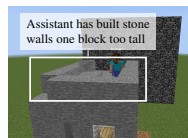 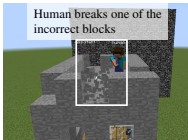 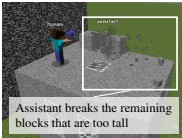

**Learning from corrections:** the assistant builds the walls too tall, but when the human breaks one of the blocks it learns the correct height and breaks the others.

Figure 1: We develop an AI assistant that helps users build houses in Minecraft using *assistance games*, an alternative to reinforcement learning from human feedback (RLHF). Our assistant helps real human players build a variety of goal houses it has never seen during training. It displays emergent behaviors like understanding pragmatic communication and learning from corrections.

## 1. Introduction

The pipeline of pretraining, supervised fine-tuning (SFT), and reinforcement learning from human feedback (RLHF) or its variants has become the dominant paradigm for training general AI assistants. RLHF involves fine-tuning pretrained foundation models to take actions (i.e., produce responses) that are preferred by human annotators according to criteria like helpfulness and harmlessness (Bai et al., 2022). However, RLHF-trained assistants have a number of drawbacks. Annotators can be fooled into giving positive feedback for unhelpful actions, incentivizing deceptive or manipulative assistant behavior (Lang et al., 2024; Williams et al., 2025). Furthermore, RLHF does not encourage models to maintain *uncertainty* about a user's goals; the objective of producing highly rated single-turn responses discourages the assistant from asking clarifying questions or hedging its

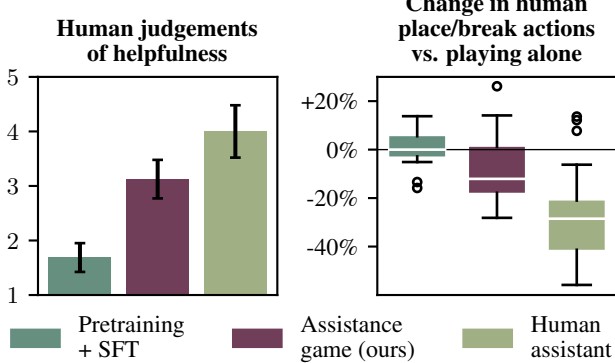

Figure 2: In a human study, we find that our assistant significantly reduces the number of actions taken by participants when compared to building without an assistant. Our assistance game-based assistant is judged as considerably more helpful than one trained with a pretraining and supervised fine-tuning (SFT) pipeline, and is rated nearly as helpful as an expert human assistant. Error bars on the left plot indicate 90% confidence intervals; box plots on the right indicate the median, quartiles, range, and outliers.

responses (Shani et al., 2024). Non-chatbot AI assistants like GitHub Copilot (Chen et al., 2021) suffer similar problems: when a coding task is ambiguous, Copilot cannot ask for clarification. Furthermore, autocomplete assistants like Copilot do not take into account the collaborative nature of assistance—an AI assistant's actions should *complement* its user's actions rather than merely predicting or replacing them.

An alternative paradigm for training AI assistants is *assistance games* (Fern et al., 2014; Hadfield-Menell et al., 2016; Shah et al., 2020). Assistance games avoid the aforementioned drawbacks of RLHF by explicitly accounting for both the interactive nature of assistance and uncertainty about the user's goal. In particular, an assistance game is a two-player game in which an assistant and a user take actions in a shared environment (Figure 3b). The two agents share a reward function, but crucially the assistant is initially uncertain about it. Assistance games remove incentives for deception since the assistant's performance depends on the true latent reward function, rather than human feedback. They also incentivize the assistant to interact with the user to resolve its uncertainty. Finally, solving assistance games results in assistants whose actions complement the user's actions to achieve optimal joint performance. In the conclusion (Section 6), we envision a recipe for applying assistance games to LLM post-training to replace RLHF.

Given the advantages of assistance games, why do they remain a poorly studied method for training AI assistants? Assistance games have been used to solve very toy problems, but have been largely dismissed in complex settings due to

seemingly insurmountable challenges. First, the AI assistant must maintain uncertainty over reward functions and make decisions under that uncertainty, which is considered computationally intractable (Papadimitriou & Tsitsiklis, 1987; Madani et al., 2003).

Second, unlike RLHF, solving assistance games requires a *human model* that can predict a human's response to AI actions. An accurate human model is essential to produce a value-aligned AI (Fisac et al., 2020); if the AI assistant fails to understand human communication strategies, it could perform poorly with real humans (Carroll et al., 2019). Past work on assistance games has used RL- or planning-based human models (Woodward et al., 2020; Zhi-Xuan et al., 2024), which can differ significantly from real human behavior.

We tackle these challenges and show that complex assistance games *can* be tractably solved. To do so, we introduce a new assistance game benchmark, the Minecraft Building Assistance Game (MBAG), in which an AI assistant helps a human build a goal structure in a Minecraft-based environment without prior knowledge of the goal (Figure 1). Creating an effective assistant in MBAG is a major challenge because the distribution over goal structures is highly complex, and the number of possible goals is far larger than in prior work (over $10^{400}$, compared to less than 20); the state and action spaces are significantly larger as well.

Using MBAG, we investigate whether deep reinforcement learning (RL) algorithms are capable of solving assistance games. We find that PPO, a popular model-free RL algorithm, can easily build known goal houses in MBAG; however, it struggles to help when the goal structure is unknown. We believe PPO fails because it requires learning both how to *predict* the goal and *act* based on its predictions simultaneously from high variance feedback.

Thus, to better solve assistance games, we introduce a new algorithm called **AssistanceZero** that separates prediction and action by extending AlphaZero (Silver et al., 2017). Similarly to AlphaZero, AssistanceZero combines Monte Carlo tree search (MCTS) with a neural network to choose actions. AssistanceZero employs a neural network with additional heads that predict rewards and human actions, which are used by MCTS to effectively plan under uncertainty (Figure 4). AssistanceZero results in much more effective assistants than PPO (Table 1). We also tackle the second challenge of solving assistance games by exploring how to develop effective human models that produce helpful assistants. Interestingly, we find that the best human models in MBAG also combine MCTS with imitation learning, a method known as piKL (Jacob et al., 2022).

We compare policies trained via an assistance game to those trained with other approaches, such as a pipeline

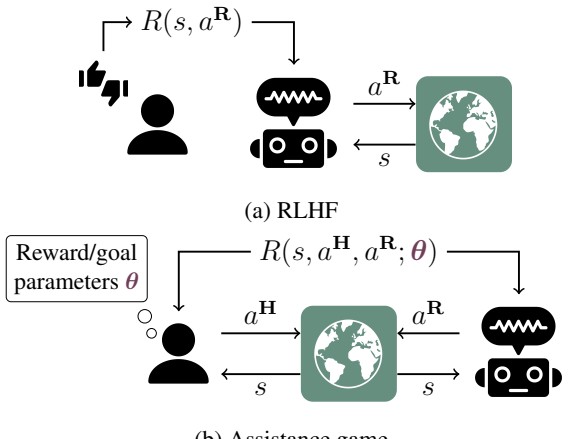

(a) RLHF

(b) Assistance game

Figure 3: Assistance games are an alternative paradigm to RLHF for developing helpful and harmless AI assistants. In RLHF (top), an assistant policy is trained to take in the environment state (e.g., human chat messages) and produce an action (e.g., a response message). The assistant policy is trained to maximize a reward function which is learned from human feedback. In contrast, in assistance games (bottom), the human is assumed to be another agent acting in the same environment as the assistant, rather than an exogenous source of feedback. The human and assistant share a reward function, but it depends on *reward parameters* that are initially known only to the human.

analogous to pretraining and SFT. In MBAG, we find that AssistanceZero-trained assistants greatly outperform those trained with pretraining+SFT or other approaches, both with our best human model (Table 3) and with real humans (Figure 2). The AssistanceZero assistant displays many helpful emergent behaviors, such as adapting based on corrections (Figure 1). Overall, our results suggest that assistance games are tractable to scale and can be a superior framework for training helpful assistants in challenging environments. We believe our approach can be extended to creating assistants for a range of real-world settings, such as AI pair programmers that help solve coding tasks.

Our contributions may be summarized as: (1) we overcome the difficulties of solving assistance games by proposing AssistanceZero, a new model-based RL algorithm; (2) we show that assistant policies trained via assistance games outperform those trained via other assistance paradigms, both in simulation and with real humans; (3) we introduce MBAG, a benchmark for assistance games with exponentially more goals than in prior work; and, (4) we investigate approaches to human modeling and determine the most effective human models for solving assistance games.

## 2. Background and related work

We begin by introducing the assistance game formalism and surveying related work. An assistance game is a Markov game in which two players, the human **H** and the assistant **R**, interact to optimize a shared reward function. It consists of a state space $\mathcal{S}$, action spaces $\mathcal{A}^{\mathbf{H}}$ and $\mathcal{A}^{\mathbf{R}}$ for the human and assistant, a set of possible reward parameters $\Theta$, and a discount factor $\gamma \in [0, 1]$. The reward parameters can represent any information that encodes the shared goal or task; for example, in a coding task, they could consist of a set of test cases that the solution should pass. Reward parameters and an initial state are sampled from a predefined distribution $p(s_1, \theta)$. At each timestep $t = 1, \ldots, T$, both agents select actions $a_t^{\mathbf{H}} \in \mathcal{A}^{\mathbf{H}}, a_t^{\mathbf{R}} \in \mathcal{A}^{\mathbf{R}}$; receive shared reward $R(s_t, a_t^{\mathbf{H}}, a_t^{\mathbf{R}}; \theta)$; and the environment transitions to state $s_{t+1}$ according to a transition distribution $p(s_{t+1} \mid s_t, a_t^{\mathbf{H}}, a_t^{\mathbf{R}})$.

A human policy $\pi_{\mathbf{H}} : \mathcal{S} \times \Theta \to \Delta(\mathcal{A}^{\mathbf{H}})$ defines a distribution over actions $\pi_{\mathbf{H}}(a^{\mathbf{H}} \mid s, \theta)$ given an environment state and reward parameters. An assistant policy $\pi_{\mathbf{R}} : (\mathcal{S} \times \mathcal{A}^{\mathbf{H}} \times \mathcal{A}^{\mathbf{R}})^* \times \mathcal{S} \to \Delta(\mathcal{A}^{\mathbf{R}})$ defines a distribution over actions $\pi_{\mathbf{R}}(a_t^{\mathbf{R}} \mid h_t)$ conditioned on the state-action history up until the current timestep: $h_t = (s_1, a_1^{\mathbf{H}}, a_1^{\mathbf{R}}, \ldots, s_{t-1}, a_{t-1}^{\mathbf{H}}, a_{t-1}^{\mathbf{R}}, s_t)$. Note that the assistant policy is *not* conditioned on the reward parameters since it cannot observe them. While in general a human policy might also depend on $h_t$, for simplicity we assume that $\pi_{\mathbf{H}}$ is only conditioned on $(s, \theta)$; previous results show there is an optimal human policy conditioned only on $(s, \theta)$ (Hadfield-Menell et al., 2016). Given a pair of policies $(\pi_{\mathbf{H}}, \pi_{\mathbf{R}})$, we can define their joint expected return as

$$J(\pi_{\mathbf{H}}, \pi_{\mathbf{R}}) = \mathbb{E}\left[\sum_{t=1}^{T} \gamma^{t-1} R(s_t, a_t^{\mathbf{H}}, a_t^{\mathbf{R}}; \theta)\right],$$

the expected discounted sum of their shared reward, where $(s_1, \theta) \sim p(s_1, \theta)$; $a_t^{\mathbf{H}} \sim \pi_{\mathbf{H}}(a^{\mathbf{H}} \mid s_t, \theta)$; $a_t^{\mathbf{R}} \sim \pi_{\mathbf{R}}(a^{\mathbf{R}} \mid h_t)$; and $s_{t+1} \sim p(s_{t+1} \mid s_t, a_t^{\mathbf{H}}, a_t^{\mathbf{R}})$. For a fixed human policy $\pi_{\mathbf{H}}$, we define a *best response* to it as an assistant policy $\pi_{\mathbf{R}}$ that maximizes $J(\pi_{\mathbf{H}}, \pi_{\mathbf{R}})$.

**Related work** Assistance games were introduced by Fern et al. (2014) and Hadfield-Menell et al. (2016) under the names "hidden-goal MDPs" and "cooperative inverse reinforcement learning." A few prior works have explored small-scale assistance games (Dragan & Srinivasa, 2013; Javdani et al., 2015; Malik et al., 2018; Fisac et al., 2020; Woodward et al., 2020; Zhi-Xuan et al., 2024) with around ten or fewer discrete reward parameters, small 2D gridworlds, and unrealistic goals, such as collecting lemons or gemstones. We aim to scale assistance games to much larger structured reward parameter spaces, similar to the goals real humans have when interacting with assistants; in our environment $|\Theta| \approx 10^{400}$.

Our approach to solving assistance games builds on tech-

niques for scalably solving games (Silver et al., 2017; Brown et al., 2020; Hu et al., 2021a), modeling human behavior (Carroll et al., 2019; Laidlaw & Dragan, 2021; Yang et al., 2022; Jacob et al., 2022), and training effective collaborative agents (Stone et al., 2010; Hu et al., 2020; Treutlein et al., 2021; Strouse et al., 2021; Hu et al., 2021b; Bakhtin et al., 2023). Minecraft and Minecraft-like environments have been previously used as testbeds for assistance and collaboration (Szlam et al., 2019; Gray et al., 2019; Bara et al., 2021; Skrynnik et al., 2022; Kiseleva et al., 2022; Zholus et al., 2022; Mehta et al., 2024) as well as for general interactive learning (Kanervisto et al., 2021; Baker et al., 2022; Fan et al., 2022; Milani et al., 2023; Wang et al., 2024).

## 3. The Minecraft Building Assistance Game

To investigate how to solve complex assistance games, we introduce the Minecraft Building Assistance Game (MBAG). When designing MBAG, we aimed to satisfy a few desiderata to make it a useful environment for studying assistance games more broadly. First, the distribution over reward parameters $p(\theta)$ should be complex but structured, similar to human preferences in other domains. As described in the related work, most past work on assistance games has considered only a small number of possible reward functions. Second, there should be a variety of ways for the assistant to help the human that require varying amounts of information about the reward function. Finally, the environment should be tractable for academic labs to train RL agents, making it feasible to empirically study more complex assistance games. In the remainder of this section, we describe the structure and implementation of MBAG.

A state in MBAG consists of a 3-dimensional grid of blocks, player locations within the grid, and player inventories. Each location in the grid can be one of ten block types, including air; we use an $11 \times 10 \times 10$ grid for our experiments. Each agent, or player, can be at any unoccupied discrete location within the 3-dimensional grid. The action space consists of a no-op, moving in one of the six cardinal directions, placing a block, or breaking a block. Place and break actions are parameterized by a location, and place actions are also parameterized by a block type. This means that in the $11 \times 10 \times 10$ environment there are over 20,000 possible actions. The players can only reach a limited distance to break or place blocks and many actions are invalid given the current state (e.g., it is impossible to break an air block); thus, usually a small subset of all actions are valid.

The reward parameters $\theta$ consist of a goal grid of blocks. To assign rewards for human and assistant actions, we use the edit distance $d(s, \theta)$ between the current state $s$ and the goal $\theta$, i.e., the minimum number of place and break block actions necessary to transform $s$ to the goal. The reward function $R(s, a^{\mathbf{H}}, a^{\mathbf{R}}; \theta) = d(s', \theta) - d(s, \theta)$ is the

| Assistant | Overall goal % | Human actions | Assistant goal % |
|---|---|---|---|
| PPO baseline | $71.6 \pm 1.0$ | $203 \pm 3$ | $0.0 \pm 0.8$ |
| $-$ LSTM | $72.4 \pm 0.9$ | $200 \pm 3$ | $2.2 \pm 0.7$ |
| $+$ rew. engineering | $74.0 \pm 0.9$ | $200 \pm 3$ | $3.5 \pm 0.7$ |
| $+$ aux. loss | $74.1 \pm 0.9$ | $191 \pm 3$ | $7.2 \pm 1.0$ |
| AssistanceZero | $\mathbf{79.8 \pm 0.9}$ | $\mathbf{158 \pm 3}$ | $27.0 \pm 1.5$ |
| $-$ test-time MCTS | $\mathbf{80.2 \pm 0.9}$ | $\mathbf{158 \pm 3}$ | $27.3 \pm 1.3$ |
| *Human alone* | $70.8 \pm 1.0$ | $200 \pm 3$ | — |

Table 1: Our proposed algorithm AssistanceZero produces more effective assistants for a fixed human model compared to a carefully tuned PPO implementation. We evaluate how well assistant policies perform with an imitation learning-based human model at building goal structures not seen during training. See Section 4 for details.

difference in edit distance before and after the assistant and human actions. This means that correct (incorrect) place or break actions give a reward of +1 (-1).

At the start of an episode, the goal is sampled from a dataset of houses based on the CraftAssist dataset (Gray et al., 2019). We maintain separate train and test datasets to evaluate generalization. While the human agent can observe the goal, it is not visible to the assistant. MBAG satisfies our first desideratum because there is an exponentially large number of possible goals (on the order of $10^{400}$), making the goal distribution much more complex than prior studies of assistance games. However, due to the structured nature of the houses, the assistant can still infer information about the goals from human interaction. MBAG also satisfies the second desideratum because some assistant strategies, like digging a foundation, require very little knowledge of the goal. On the other hand, adding final decorations requires specific information. For more details about the MBAG environment, see Appendix B.

## 4. Solving assistance games with AssistanceZero

Using MBAG, we first examine how to solve the complex problem of sequential decision-making under uncertainty posed by assistance games. We begin by assuming we have a fixed human policy $\pi_{\mathbf{H}}(a^{\mathbf{H}} \mid s, \theta)$ and study how to find a best response assistant policy. For now, we use a human model $\pi_{\mathbf{H}}$ based on imitation learning; see Section 4.3 for more details about our approach to human modeling.

### 4.1. PPO fails to solve assistance games

Shah et al. (2020) show that finding a best response to a fixed human policy in an assistance game is equivalent to solving a single-agent partially observable Markov decision

process (POMDP); we call this an *assistance POMDP*. An effective tool to solve many POMDPs is model-free deep RL, which leverages the generalization capabilities of deep neural networks to perform well in environments that are intractable to solve via other methods like dynamic programming or planning (Ni et al., 2022). In particular, proximal policy optimization (PPO) (Schulman et al., 2017) with a recurrent policy network has shown promise in a variety of partially observable and multi-agent settings (OpenAI et al., 2019; Yu et al., 2022).

We use PPO to train assistant policies in MBAG through a standard model-free RL loop. PPO collects a set of rollouts from several environments in parallel; human actions are sampled from the fixed human model $\pi_{\mathbf{H}}$, and assistant actions are sampled from the current assistant policy $\pi_{\mathbf{R}}$, which is parameterized as a convolutional neural network (Hochreiter & Schmidhuber, 1997). At the beginning of each training episode, a goal structure $\theta$ is randomly sampled from the training dataset $\mathcal{D}_{\text{train}}$. Then, PPO optimizes the assistant policy's parameters using a surrogate loss function which aims to increase the policy's reward.

To test our PPO assistant policy, we evaluate it with the same imitation learning-based human model over 1,000 episodes with goal structures from our test set $\mathcal{D}_{\text{test}}$. We collect three performance metrics: the *average percentage of the goal structure that is completed*, the *total number of place and break blocks taken by the human*, and the *percentage of the total goal structure built by the assistant*. We also evaluate the human model playing alone. Compared to this baseline, ideally the human model-assistant pair should achieve an equal or higher goal percentage while requiring fewer human actions. See Appendix E for the full details of our training and evaluation setup.

Unfortunately, we found that PPO struggles in MBAG. An assistant trained with recurrent PPO does not help the human model at all (first row of Table 1). Surprisingly, non-recurrent PPO slightly outperforms recurrent PPO (second row). We believe this setting is challenging for PPO due to the high variance of the reward signal it uses for learning. Since the reward function is shared, the reward depends not only on the assistant's actions, but also on those of the human model, which the assistant can only control indirectly. Furthermore, since the assistant is uncertain about the goal structure, even taking an action that is helpful in expectation given the observation history will sometimes result in negative reward. The sequential and long-horizon nature of the task exacerbates these issues, further increasing the noise in the reward-to-go signal that PPO seeks to optimize.

As a result, the most discernible signal PPO receives early in training is that place and break actions tend to be incorrect, incurring negative reward. Thus, the assistant policy converges to building little to nothing. To decrease the noise

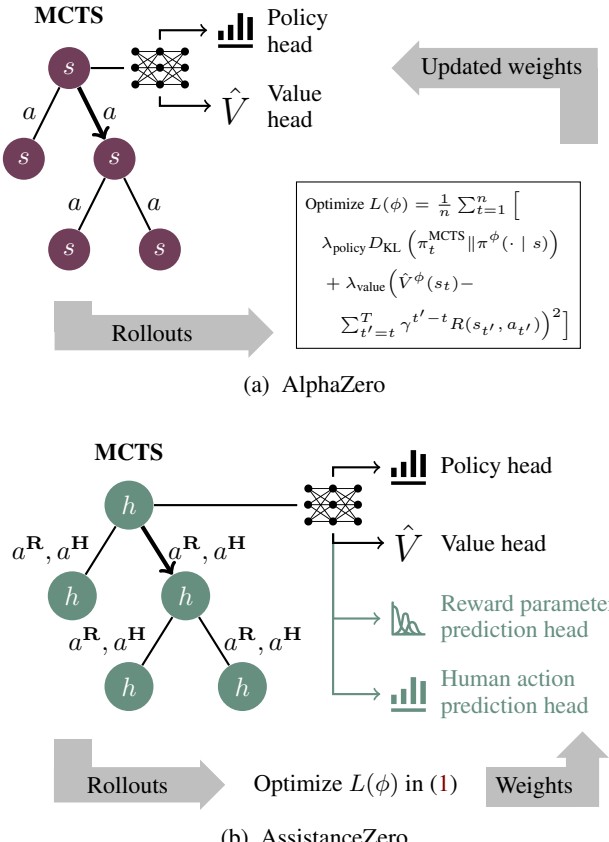

(a) AlphaZero

(b) AssistanceZero

Figure 4: AssistanceZero (bottom) extends AlphaZero (top) to solve assistance games. While AlphaZero requires access to the transition and reward functions to run MCTS, in assistance games the rewards and human actions depend on the reward parameters $\theta$, which are not visible to the assistant. AssistanceZero learns to predict the reward parameters and human actions from rollouts, enabling it to plan with MCTS and train an effective assistant policy.

in the reward signal and incentivize the assistant to act more, we explore training the assistant based on only the reward from its own actions[1]. We also experiment with adding an auxiliary loss term to encourage placing the correct blocks. These slightly increase the percentage of the goal built by the assistant-human model pair while reducing or maintaining the number of human model actions (third and fourth of Table 1). However, they are still only barely helpful. Thus, to tractably solve complex assistance games such as MBAG, we turn to an alternative approach.

## 4.2. AssistanceZero

Given the failure of PPO to train effective assistant policies in MBAG, we propose a different algorithm for solving as-

---

[1]This no longer solves the assistance game and could be dangerous; the assistant may be incentivized to prevent the human from taking actions so that it can take them instead.

sistance POMDPs: AssistanceZero. We hypothesize that PPO struggles because the reward signal is very noisy, and it must learn to both *predict* the goal structure and *act* based on its predictions from this noisy signal. Thus, we design AssistanceZero to *separate goal prediction and action selection* by learning a goal predictor and then using it for planning. Specifically, AssistanceZero is an extension of AlphaZero, a deep RL algorithm that has achieved super-human performance in complex competitive games like Go and chess (Silver et al., 2017). Like AlphaZero, AssistanceZero chooses actions using a variant of Monte Carlo tree search (MCTS) (Kocsis & Szepesvári, 2006). MCTS builds a search tree by simulating the results of taking different sequences of actions from the current state. However, it requires knowledge of both the *reward* and the *next state* resulting from an action, neither of which is known in an assistance POMDP: the next state depends on the human's action, and the reward $R(s, a^{\mathbf{H}}, a^{\mathbf{R}}; \theta)$ depends on the reward parameters $\theta$ which are not visible to the assistant.

To overcome these challenges, AssistanceZero employs a recurrent neural network with parameters $\phi$ that takes as input a state-action history $h$ and has four heads: a *policy* head $\pi^\phi(a^{\mathbf{R}} \mid h)$, a *value* head $\hat{V}^\phi(h)$, a *reward parameter prediction* head $\hat{p}^\phi(\theta \mid h)$, and a *human action prediction* head $\hat{p}^\phi(a^{\mathbf{H}} \mid h)$. The policy and value heads select actions and evaluate the value of states, respectively, similarly to the policy and value networks in AlphaZero. The reward parameter and human action prediction heads predict distributions over $\theta$ and $a^{\mathbf{H}}$ so that MCTS can estimate the reward and next state given a selected action. Concretely, in MBAG, the reward parameter head predicts a probability distribution over the possible block types at each location in the world.

Similar to PPO, we train the AssistanceZero network by collecting rollouts in several parallel environments, selecting assistant actions using MCTS with the current network parameters. Then, the four heads are trained using separate loss terms. As in AlphaZero, the policy head is updated to minimize the KL divergence towards the policy output from MCTS, and the value head to minimize the squared error with the reward-to-go. The reward parameter and human action prediction heads are trained with negative log-likelihood loss to predict $\theta$ and $a^{\mathbf{H}}$, respectively. We found that the reward parameter prediction head is prone to overfitting to the most recently seen goal structures, so we additionally include a KL divergence term from the current prediction $\hat{p}^\phi(\theta \mid h_t)$ to the predictions made when $h_t$ was originally sampled, which we denote as $\hat{p}_t(\theta)$. The full AssistanceZero loss can be written for a trajectory of $n$

timesteps as

$$
\begin{aligned}
L(\phi) = \frac{1}{n} \sum_{t=1}^{n} \Big[ & \lambda_{\text{policy}} D_{\text{KL}} \left( \pi_t^{\text{MCTS}} \| \pi^\phi(\cdot \mid h_t) \right) \\
& + \lambda_{\text{value}} \left( \hat{V}^\phi(h_t) - \sum_{t'=t}^{T} \gamma^{t'-t} R(s_{t'}, a_{t'}^{\mathbf{H}}, a_{t'}^{\mathbf{R}}; \theta) \right)^2 \\
& - \lambda_{\text{reward}} \log \hat{p}^\phi(\theta \mid h_t) + \lambda_{\text{prev-rew}} D_{\text{KL}} \left( \hat{p}^\phi(\theta \mid h_t) \| \hat{p}_t(\theta) \right) \\
& - \lambda_{\text{action}} \log \hat{p}^\phi(a_t^{\mathbf{H}} \mid h_t) \Big],
\end{aligned} \tag{1}
$$

where $\lambda_{\text{policy}}, \lambda_{\text{value}}, \lambda_{\text{reward}}, \lambda_{\text{prev-rew}}$, and $\lambda_{\text{action}}$ are weights that trade off the five loss terms, and $\pi_t^{\text{MCTS}}$ refers to the action distribution output by MCTS at timestep $t$. After a few epochs of gradient descent on $L(\phi)$ over the collected episodes, AssistanceZero collects new episodes by running MCTS with the updated network parameters and repeats the process. The technique of learning an approximate belief distribution over the reward parameters $\theta$ from rollouts is similar to learned belief search (Hu et al., 2021a). The variant of MCTS employed by AssistanceZero is also similar to POMCP (Silver & Veness, 2010), a variant of MCTS for POMDPs, except that we use a learned model of the environment. AssistanceZero is also related to model-based extensions of AlphaZero like MuZero (Schrittwieser et al., 2020); however, MuZero assumes full observability and that the next state is deterministic, which is not the case in assistance games. See Appendix A for a full description of AssistanceZero and our variant of MCTS.

We train and evaluate AssistanceZero assistant policies using the same setup as the PPO assistants; the results are shown in the bottom row of Table 1. Our AssistanceZero assistant significantly outperforms PPO-based assistants across all metrics, increasing the percentage of the goal completed by building 27% of the structure while reducing the number of human model actions by 42. To ensure a fair comparison, we also evaluate AssistanceZero without MCTS at test-time, using only the policy head to select actions. This does not reduce the assistant's performance, demonstrating that AssistanceZero does not outperform PPO simply because it uses additional test-time compute.

### 4.3. Choosing a human model

While we have shown that AssistanceZero can train assistants that perform well with a fixed human model, it remains unclear how to obtain a good human model in the first place. Ideally, an assistant policy should perform well not only with the human model it was trained with, but with real humans. We explore a number of approaches from the human-AI interaction literature for developing human models in MBAG, including reward-based and data-based models.

Reward-based human models assume that humans choose actions approximately optimally to maximize their reward function. We use deep RL to train two reward-based models to build goal structures by themselves. For one model, we use PPO with an entropy coefficient, which approximates

| Human model | Cross entropy | | Goal % after X min | | | |
|---|---|---|---|---|---|---|
| | Alone | w/ asst. | 3 | 5 | 10 | 20 |
| PPO | 12.23 | 12.24 | 79 | 96 | 99 | 100 |
| AlphaZero | 6.85 | 6.52 | 82 | 97 | 100 | 100 |
| BC-alone | 2.11 | 2.15 | 8 | 13 | 30 | 58 |
| BC-with-asst. | 2.13 | 2.06 | 10 | 18 | 40 | 71 |
| BC-combined | **1.89** | **1.99** | 9 | 17 | 41 | 71 |
| piKL-alone | 2.18 | 2.37 | 25 | 40 | 66 | 82 |
| piKL-with-asst. | 2.25 | 2.29 | 26 | 42 | 74 | 92 |
| piKL-combined | 1.98 | 2.20 | 26 | 44 | 75 | 91 |
| Humans subjs. | — | — | 25 | 42 | 80 | 95 |

Table 2: We evaluate eight human models based on their cross entropy with the actions of real humans (playing either with or without an assistant) and how well they perform at building goal structures alone compared to human subjects. We find that the reward-based human models, PPO and AlphaZero, are poor predictors of human actions and build houses faster than human subjects. BC models predict human actions well but build houses more slowly than human subjects. Finally, piKL models, which combine the BC models with planning, predict human actions well and build houses at a similar rate to human subjects. The most accurate BC and piKL models are trained on the combined human-alone and human-with-assistant data.

Boltzmann rationality, a common noisily-optimal model of human behavior (Luce, 1959; 1977; Ziebart et al., 2010). We train the other model using AlphaZero.

Next, we train a series of data-based human models using behavior cloning (BC), which predicts actions from states using supervised learning. For the training dataset, we record 18 episodes in MBAG of five human subjects building houses randomly selected from $\mathcal{D}_{\text{train}}$. In half of these episodes the human builds alone and in the other half an experienced Minecraft player acts as an assistant. We display the goal structure to subjects as a transparent blueprint overlaying the normal Minecraft game, while keeping it hidden from the human assistant. Using BC, we train three human models: one on the data where the subject played alone (BC-alone), one on the subset played with the assistant (BC-with-assistant), and one on the whole dataset (BC-combined); see Appendix E.1 for details. While our formal definition of assistance games assumes that the human model is Markov, we find that a recurrent, history-based BC model is more predictive of human actions than a Markov policy. Besides capturing the non-Markovian behavior of individual humans, a recurrent human model can also implicitly model a mixture of human policies. This allows a single recurrent model to potentially capture the variance in the skill levels of real humans.

Some recent work has proposed combining reward-based

and data-based human models (Cornelisse & Vinitsky, 2024). To explore this type of human modeling, we implement piKL (Jacob et al., 2022), which uses MCTS with an imitation-learned prior policy to select actions that maximize reward but are also human-like. We experiment with piKL models based on each of our three BC models.

We evaluate all eight human models according to prediction accuracy, performance alone, and efficacy for training assistants. To measure prediction performance, we calculate the cross entropy of each model on human data; for the BC and piKL models, we use cross-validation. We also evaluate each human model building 1,000 goal structures alone to determine how well it performs compared to our human subjects. Finally, for each human model, we train an assistant with AssistanceZero and then evaluate the assistant policy with every other human model for 100 episodes. This helps determine if a human model leads to an assistant that generalizes well to other human models. See Appendix D.1 for more details on our human model training and evaluation.

The results of our human model evaluations are shown in Table 2 and Figure 9. Similarly to past work (Carroll et al., 2019; Laidlaw & Dragan, 2021; Bakhtin et al., 2021), we find that pure reward-based models are poor predictors of human actions. Both the PPO and AlphaZero human models have very high cross entropy with real human actions and build goal structures much more quickly than human subjects. The BC human models have considerably lower cross entropy, with the lowest cross entropy achieved by the BC model trained on the combined BC dataset. However, they also seem to suffer from compounding errors, i.e., small prediction errors accumulating over time (Ross et al., 2011), and thus build less of the goal structure than real humans. The piKL models are slightly less predictive in terms of cross entropy but closely match human performance.

The results of training AssistanceZero assistants with one human model and testing with another are shown in Figure 9. We evaluate each assistant-human model pair based on both the average goal percentage completed and the mean number of human actions. Compared to the human models building alone, in most cases assistants are able to maintain or increase the goal percentage while decreasing the number of human actions, demonstrating their effectiveness. Overall, the piKL human models seem to produce the best assistants according to both metrics. We chose to use the AssistanceZero assistant trained with the **piKL-combined** human model for the remainder of our experiments. It achieves low cross entropy on human data, similar performance by itself to humans alone, and produces an assistant that generalizes to other human models.

| Assistant training | Overall goal % | Human actions | Assistant goal % |
|---|---|---|---|
| Pretraining | $89.8 \pm 0.7$ | $240 \pm 4$ | $2.3 \pm 0.5$ |
| SFT | $90.4 \pm 0.7$ | $241 \pm 4$ | $2.9 \pm 0.3$ |
| Assistance game | $\mathbf{92.6 \pm 2.4}$ | $\mathbf{179 \pm 11}$ | $\mathbf{26.0 \pm 3.3}$ |
| *Hum. model alone* | $90.0 \pm 0.8$ | $245 \pm 4$ | — |

Table 3: We compare three approaches to building assistants in our MBAG benchmark: pretraining, which is analogous to autocomplete-based assistants like GitHub Copilot; SFT, which is analogous to the first stage of RLHF; and assistance games. We evaluate the assistant policy trained with each approach based on the same metrics as Table 1. The policy based on assistance games outperforms the others in all metrics, building around a quarter of the goal structure itself and allowing the human to take many fewer actions.

## 5. Comparing assistance paradigms

Given our complete recipe for training an assistant in MBAG via assistance games—fixing a piKL policy for the human model and then using AssistanceZero to solve the resulting assistance POMDP—we now compare assistance games to other paradigms for training AI assistants. In particular, we develop pipelines for training MBAG assistants analogous to those used by GitHub Copilot/OpenAI Codex (Chen et al., 2021) and the supervised fine-tuning (SFT) stage of RLHF (Bai et al., 2022; Ouyang et al., 2022), since these are two dominant paradigms for training current AI assistants. We compare the resulting policies to our AssistanceZero-trained assistant.

Both RLHF and Codex begin with pretrained language models, which allows them to learn useful representations and to be able to predict human actions. One way to view the pretraining data is that it consists of humans solving a variety of tasks. For example, Codex was trained on GitHub, and files in GitHub can be viewed as human demonstrations of solving various programming tasks. Thus, in MBAG, we analogously generate a pretraining corpus by using the BC-combined human model to generate 10,000 episodes where it builds randomly selected goal structures from our training set $\mathcal{D}_{\text{train}}$. We then remove information about the goal structure from the observations and train a recurrent neural network on the resulting dataset, which we refer to as the **pretrained model**. Similarly to language or code models, this model can predict human actions without goal information and has learned representations that allow it to understand the structure of human goals. By sampling actions from the pretrained model at a low temperature, we obtain an assistant similar to GitHub Copilot: it acts to build the goal structure when it is highly confident about which actions the human will take, and does not take actions when it is unconfident.

We further train the pretrained model using supervised fine-tuning (SFT), the first stage of RLHF. For SFT, we use data of a human expert acting as the assistant from the same data collection sessions used to train the BC-with-assistant human models. We fine-tune the pretrained model to imitate the human assistant, similar to how LLMs are trained to imitate human-written assistant responses during the SFT stage of RLHF. We use a grid search over 540 hyperparameter combinations to find the best combination of learning rate, training epochs, data augmentation, and dropout for the **SFT policy**; see Appendix E.3.1 for details.

We do not directly compare to a full RLHF baseline because it is not easily applicable to the MBAG environment. RLHF is usually formulated as a single-agent problem (Christiano et al., 2017; Ouyang et al., 2022), so the additional human agent in MBAG would make it difficult to apply standard techniques. Furthermore, in LLMs, RLHF is applied to only a single step of interaction between the assistant and the user, i.e., the comparison data used by RLHF uses conversations which only differ in the last assistant message. In MBAG, the equivalent would be to compare single assistant actions taken in response to a given history of human and assistant actions. However, it may be quite difficult to judge assistant actions in isolation; for instance, more than half of assistant actions are usually movement, and it is unclear how to judge the relative usefulness of say, moving left versus up. For these reasons, we decided to only compare to an SFT baseline, especially since SFT alone for LLMs can often achieve performance close to that of RLHF (Zhou et al., 2023).

**Evaluation with human models** We compare the pretrained and SFT models to our assistance game-based policy in Table 3. We evaluate each with the piKL-combined human model over 1,000 episodes and report the same metrics as in Table 1. Both the pretrained and SFT policies slightly decrease the number of human actions (by around 4-5) needed to achieve a similar goal completion percentage. The SFT policy builds around 3% of the goal structure on average. In contrast, the policy trained with AssistanceZero decreases the number of human actions by around 65 while leading to a higher goal completion percentage; it builds around 26% of the goal itself.

**Human study** To validate our promising results, we measure the performance of AI assistants with real humans. We compare humans playing in four conditions: alone (no assistant), with the SFT policy, with our AssistanceZero-trained assistant, and with an expert human assistant. We use a within-subjects design where each participant builds the same house five times in a row. The first episode is used as practice to familiarize the subject with the Minecraft controls and goal structure. Then, the subject builds the house under the four conditions in a random order.

We collect both subjective and objective metrics of the assistants' helpfulness. After playing with each assistant, subjects rate its overall helpfulness, answer Likert scale agree-disagree questions about the assistant (e.g., whether it understood their intentions), and provide free-response comments. We also measure the number of actions taken by the human subject to complete the goal structure with an assistant, normalized by dividing by the number of actions needed for the subject to complete the goal alone.

An overview of the human study results are shown in Figure 2, with more results in Appendix C.1. The AssistanceZero-trained assistant performs considerably better than the SFT assistant and approaches the human baseline. Participants rate the AssistanceZero assistant's helpfulness on average as $3.1 \pm 0.4$ on a 5-point scale (90% confidence interval), while the SFT assistant is rated $1.7 \pm 0.3$ and the human baseline is rated $4.0 \pm 0.5$. Also, our assistant enables participants to build the goal structure with significantly fewer place and break actions compared to building alone (one-sided t-test $p < 0.05$). Qualitatively, participants were impressed by AssistanceZero's ability to learn effectively from corrections (e.g., breaking multiple incorrect blocks after the human broke one or two of them), while noting the SFT assistant was not helpful at all. However, there is still a sizeable gap between our assistant's performance and the expert human baseline, demonstrating that MBAG is a challenging benchmark for assistance. We hope this will inspire others to develop even more effective AI assistants in MBAG and other complex, collaborative tasks.

## 6. Conclusion

We have introduced the Minecraft Building Assistance Game and used it to show how to scalably solve assistance games using AssistanceZero. Furthermore, we have found that assistants trained via assistance games outperform those trained similarly to typical LLM post-training piplines.

**Future work: LLM post-training** In the future, assistance games can be applied to LLM post-training as well. Here, we briefly outline a vision for how this could work. To build an LLM-based assistance game, one would treat the human and assistant chat messages as actions. That is, the human and assistant alternate taking actions until the human ends the conversation, with the state consisting of all previous messages. For reward parameters, one could curate a large dataset of natural language descriptions of tasks that humans might want to solve. Then, a human model could be built by prompting an LLM to act as a human solving a given task—possibly with additional fine-tuning on abundant real human chat data. To measure reward, another LLM could evaluate whether the task is completed by the end of a chat conversation. Another possibility is to build a coding-specific assistant by representing goals as sets of test cases that should be passed by writing a block of code.

By training an LLM in this assistance game to help with the initially unknown human task, it could be possible to avoid some of the pitfalls of RLHF. Because the assistant would be optimizing over multiple chat turns and under uncertainty about the goal, it would be incentivized to ask clarifying questions, especially if the tasks are complex enough that they cannot be described in one or two messages. Furthermore, because rewards would be judged by an equally powerful LLM based on the task description, there would be less incentive for deception: if an assistant fooled the human model to appear successful, it would still receive low reward from the judge. In the case of the coding assistant, if some test cases are hidden to the human, the assistant would have the incentive to look for bugs even if the human does not notice them, since the final reward is based on the hidden test cases.

We hope our work on assistance games will eventually help LLMs move beyond simply answering questions to become effective collaborators in complex, real-world tasks.

## Impact statement

Our paper aims to improve techniques for solving assistance games, which we hope may eventually be used more broadly as a paradigm for training helpful and harmless AI assistants. As we have argued, assistance games could remove incentives for deception that exist in RLHF, the dominant current techniques for building AI assistants. Furthermore, Russell (2019) argues that assistance games could form the core of a solution to the problem of controlling superintelligent AI (Bostrom, 2016). We hope our contributions will allow future work to further explore the strengths and weaknesses of assistants trained with assistance games.

## Acknowledgements

We would like to thank Micah Carroll for acting as the expert human assistant in the user study; Mark Bedaywi, Jessy Lin, and Niklas Lauffer for feedback on drafts; and Cam Allen for helpful discussions.

This work was supported by a grant from Open Philanthropy to the Center for Human-Compatible Artificial Intelligence at UC Berkeley and a grant from the National Science Foundation (NSF) Human-Centered Computing (HCC) to Professor Anca Dragan (award number 2310757). Cassidy Laidlaw is supported by a National Defense Science and Engineering Graduate (NDSEG) Fellowship and an Open Philanthropy AI Fellowship. Eli Bronstein is supported by a National Science Foundation Computer and Information Science and Engineering Graduate Fellowship (CSGrad4US).

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

# Appendix

## A. AssistanceZero details

In this appendix, we describe the full details of the AssistanceZero algorithm.

**MCTS**    To choose actions during training and deployment, AssistanceZero uses Monte Carlo tree search (MCTS). MCTS repeats a three-stage process for $N_{\text{sim}}$ simulations, adding one additional node during each simulation to a tree where nodes represent histories and branches are action pairs $(a^{\mathbf{H}}, a^{\mathbf{R}})$.

In the *selection* stage, an assistant action $a^{\mathbf{R}}$ is selected at the current history node $h$ that maximizes

$$Q(h, a^{\mathbf{R}}) + c_{\text{PUCT}}\, \pi^{\phi}(a^{\mathbf{R}} \mid h) \frac{\sqrt{\sum_{b \in \mathcal{A}^{\mathbf{R}}} N(h, b)}}{1 + N(h, a^{\mathbf{R}})}, \tag{2}$$

where $N(h, a^{\mathbf{R}})$ is the number of times action $a^{\mathbf{R}}$ has previously been selected at node $h$, $\pi^{\phi}(a^{\mathbf{R}} \mid h)$ is the output of the network's policy head, and $c_{\text{PUCT}}$ is a tunable parameter that balances exploration and exploitation. $Q(h, a^{\mathbf{R}})$ is an estimate of the Q-value of $a^{\mathbf{R}}$; we will describe how this is calculated later. Once an assistant action is chosen, then a human action $a^{\mathbf{H}}$ is sampled according to the probabilities output by the human action predictor head $\hat{p}^{\phi}(a^{\mathbf{H}} \mid h)$. Then, the state $s'$ resulting from taking actions $(a^{\mathbf{H}}, a^{\mathbf{R}})$ is calculated and the state and actions are appended to $h$ to reach a node $h'$. The reward associated with the transition is estimated by marginalizing over the reward parameter distribution output by the reward prediction head:

$$\hat{R}(h, a^{\mathbf{H}}, a^{\mathbf{R}}) = \sum_{\theta \in \Theta} R(s, a^{\mathbf{H}}, a^{\mathbf{R}}; \theta)\, \hat{p}^{\phi}(\theta \mid h'). \tag{3}$$

Then, the selection process repeats until a node $h$ is reached which has not previously been reached.

In the *expansion stage*, the new node is input to the network to calculate the policy head outputs $\pi^{\phi}(a^{\mathbf{R}} \mid h)$, the value estimate $\hat{V}^{\phi}(h)$, the human action predictions $\hat{p}^{\phi}(a^{\mathbf{H}} \mid h)$, and the reward parameter predictions $\hat{p}^{\phi}(\theta \mid h)$. The policy outputs at the root node have Dirichlet noise applied, similarly to AlphaZero.

In the *backup stage*, the Q-values of all ancestor nodes of $h$ are recursively updated with the discounted sum of rewards along edges of the tree plus the value estimate $\hat{V}^{\phi}(h)$. As normally in MCTS, $Q(h, a^{\mathbf{R}})$ is simply the average of the Q-values estimated over all previous simulations that have taken $a^{\mathbf{R}}$ in node $h$. For actions with no visits, $Q(h, a^{\mathbf{R}})$ is set to the average of all backed-up values for node $h$:

$$Q(h, a^{\mathbf{R}}) = \frac{\sum_{b \in \mathcal{A}^{\mathbf{R}}} N(h, b) Q(h, b)}{\sum_{b \in \mathcal{A}^{\mathbf{R}}} N(h, b)} \qquad \text{if} \quad N(h, a^{\mathbf{R}}) = 0.$$

When selecting actions according to (2), we normalize Q-values by the highest and lowest value seen among all visits to that node, similarly to MuZero (Schrittwieser et al., 2020). We scale the Q-values such that the higest value seen is mapped to 1 and the lowest value seen is mapped to 0.

The resulting policy from MCTS is defined as

$$\pi^{\text{MCTS}}(a^{\mathbf{R}} \mid h) \propto N(h, a^{\mathbf{R}})^{\tau},$$

where $\tau$ is an inverse temperature parameter.

**Training procedure**    As described in Section 4.2, AssistanceZero alternates between rolling out trajectories in the environment by selecting actions with MCTS and updating the network according to the loss function in (1). Specifically, each training step consists of the following phases:

1. Run MCTS in a large number of environments in parallel to collect trajectories. Because episodes are long (1,500 timesteps), we collect only a smaller number of timesteps from each environment, which we call *fragments*. Then, all environments are paused mid-episode until the next trajectory collection phase. When an episode ends due to the completion of the goal structure or after 1,500 timesteps, a new episode begins with a newly sampled goal structure; data continues to be sampled until the required number of timesteps is reached.

2. Store the collected data in a replay buffer. Each fragment is kept as a single unit within the replay buffer to enable training recurrent policies.

3. Sample data from the replay buffer and run SGD to minimize the loss in (1), then update the networks used for sampling with the new weights.

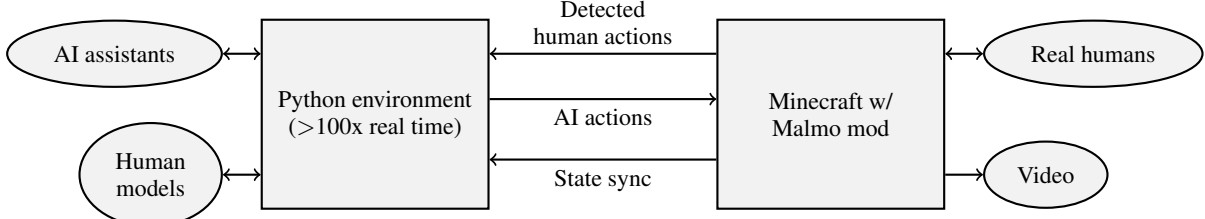

Figure 5: The architecture of the MBAG environment. The Python environment (left) can run on its own very quickly on a single CPU core, enabling efficient training for AI assistants and human models. However, it can also connect to a running Minecraft instance (right) with a custom version of the Malmo mod (Johnson et al., 2016). This enables visualizing AI policies and recording video of them; collecting data of humans playing by themselves or with each other; and, testing AI assistants with real humans.

**Lower-variance reward estimation**   There is some subtlety in the best way to estimate rewards depending on the structure of the reward function. In some environments, such as MBAG, the environment's reward function is decomposable into a component that depends only on the human's action and a component that depends only on the assistant's action:

$$R(s, a^{\mathbf{H}}, a^{\mathbf{R}}; \theta) = R^{\mathbf{H}}(s, a^{\mathbf{H}}; \theta) + R^{\mathbf{R}}(s, a^{\mathbf{R}}; \theta).$$

In this case, one can estimate the reward equivalently in expectation to (3) as

$$\hat{R}(h, a^{\mathbf{H}}, a^{\mathbf{R}}) = \sum_{\theta \in \Theta} R^{\mathbf{H}}(s, a^{\mathbf{H}}; \theta) \, \hat{p}^{\phi}(\theta \mid h') + R^{\mathbf{R}}(s, a^{\mathbf{R}}; \theta) \, \hat{p}^{\phi}(\theta \mid h). \tag{4}$$

That is, in (4) the *human's reward* is estimated based on estimated reward parameters at the *next timestep* using $h'$, while the *assistant's reward* is estimated based on the estimated reward parameters at the *current timestep* using $h$. This is preferable to (3) because the second term no longer depends on $a^{\mathbf{H}}$, which is sampled for each simulation of MCTS and thus introduces additional variance.

The reason that (4) is equivalent to (3) in expectation is that the assistant's action is independent of the reward parameters $\theta$ given the history $h$, since the assistant policy $\pi^{\mathbf{R}}(a^{\mathbf{R}} \mid h)$ only takes as input $h$ and not $\theta$. On the other hand, it is not possible to do the same to estimate the human's component of the reward, since $a^{\mathbf{H}}$ does reveal information about $\theta$.

## B. Environment details

Minecraft is typically a difficult environment to use for reinforcement learning because it is slow and resource intensive. To avoid these challenges, we implement MBAG as a "Minecraft simulator" written in a mix of pure Python and C. MBAG can be used without a running Minecraft game, allowing for training to take place more quickly and with fewer resources (MBAG can run around 100x the speed of Minecraft). However, MBAG can also interact with the Microsoft Malmo mod (Johnson et al., 2016) to allow the Python environment to sync with Minecraft. This allows policies to be visualized by watching them run in a Minecraft. It also enables human-AI play, in which human actions detected in Minecraft are translated into their equivalents in MBAG, and AI actions taken in MBAG are translated into actions in Minecraft.

We provide two versions of MBAG: one where the players must collect resources by breaking a regenerating "palette" of blocks located on one side of the environment, and one where the players have unlimited blocks. In the former version, players may also give blocks to other players; give actions are parameterized by a location, similar to place and break block actions. For the purposes of this paper, we investigate the second version with unlimited blocks; this version of the environment is more difficult to build an assistant for, since the assistant cannot simply collect resources to help the human.

### B.1. Goal structures

We base the goal structures for MBAG on the CraftAssist houses dataset, which was collected by Gray et al. (2019); they gave study participants the open-ended task of building any house in Minecraft and recorded the resulting structure. Since we require that goal structures in MBAG have a one-block gap on all sides, their dimensions can be at most $9 \times 8 \times 8$. However, many of the goal structures in the CraftAssist dataset are much larger. When houses in the dataset are no more than twice the desired dimensions, we scale them down to fit.

# C. Human study

## C.1. Full human study results

Here, we include additional results from our human study, including the participant demographics and more survey questions from the 16 subjects.

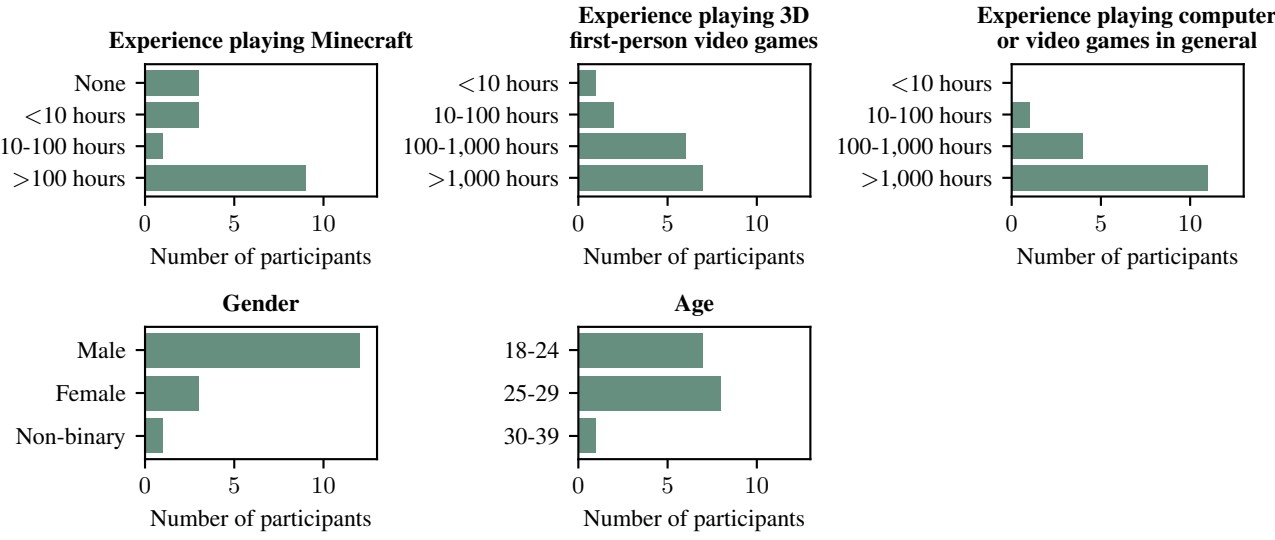

Figure 6: The demographics of the participants in our human study and their prior experience playing Minecraft and video games.

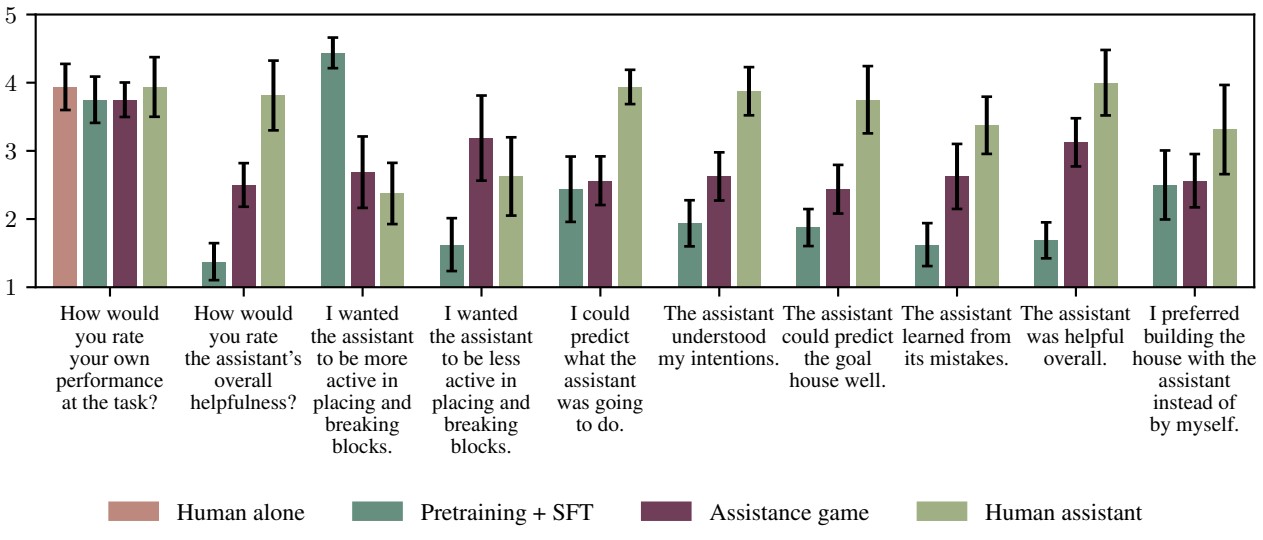

Figure 7: The full set of survey questions that participants answer after playing with each assistant. For the first two questions, participants answered with a 1-5 scale. For the remaining statements, participants answered with a 1-5 scale from "strongly disagree" to "strongly agree." The mean of the responses are shown along with 90% confidence intervals.

## C.2. Study design

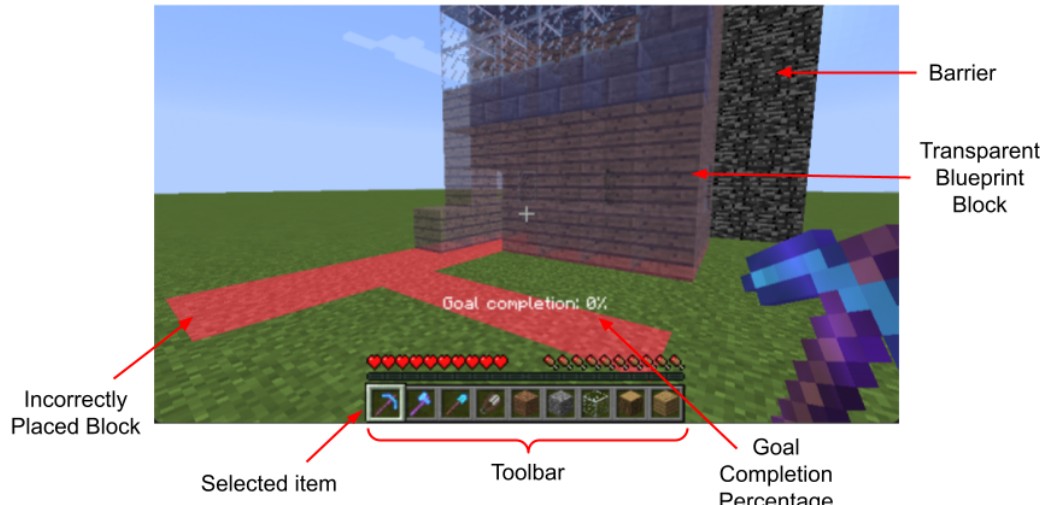

Figure 8: An example screenshot of the Minecraft game seen in the human study, which is provided to participants in the "Minecraft Guide."

We conduct the study with a total of 16 participants. To begin the study, each subject answers demographic and survey questions related to their prior experience playing Minecraft and other video games (see Figure 6 for results). Next, we describe the task of building a goal structure with an assistant where the subject can see the goal but the assistant cannot. The subject is provided with a "Minecraft Guide" describing the Minecraft mechanics, keyboard and mouse controls, and how the goal structure is visualized. There are three goal display options: the entire goal is visible as translucent goal blocks, only the currently placeable goal blocks are shown, and the goal is completely hidden (only the current world state is visible). See Figure 8 for an example screenshot.

After reading the guide, the subject plays a practice round by building a goal structure alone in order to familiarize themselves with the Minecraft environment and the goal. Next, they build the same structure in each of the four conditions—no assistant, with the SFT policy, with our AssistanceZero-trained assistant, and with an expert human assistant—in a randomly permuted order. The human assistant is an experienced Minecraft player who is not a co-author on this paper and was recruited from the same institution as the authors.

We randomly sample a unique goal structure for each participant from our test set $\mathcal{D}_{\text{test}}$. Since each subject builds their assigned goal structure five times, there may be a learning effect where the participant builds the house more quickly and efficiently for later conditions. We account for this effect by using a Latin square design. We randomly sample four permutations of the four assistance conditions, resulting in a total of 16 orders, one for each participant. The study is single-blind, meaning that subjects are not given any information about the assistant they were building with, including whether the three assistants differ from each other.

After completing the goal in each condition, the subject completes survey questions about their own and the assistant's performance. See Figure 7 for the full list of survey questions and results.

Subjects are paid $20 for their participation in the form of an Amazon gift card.

# D. Additional results

## D.1. Human modeling

### D.1.1. CROSS EVALUATION OF ASSISTANTS AND HUMAN MODELS

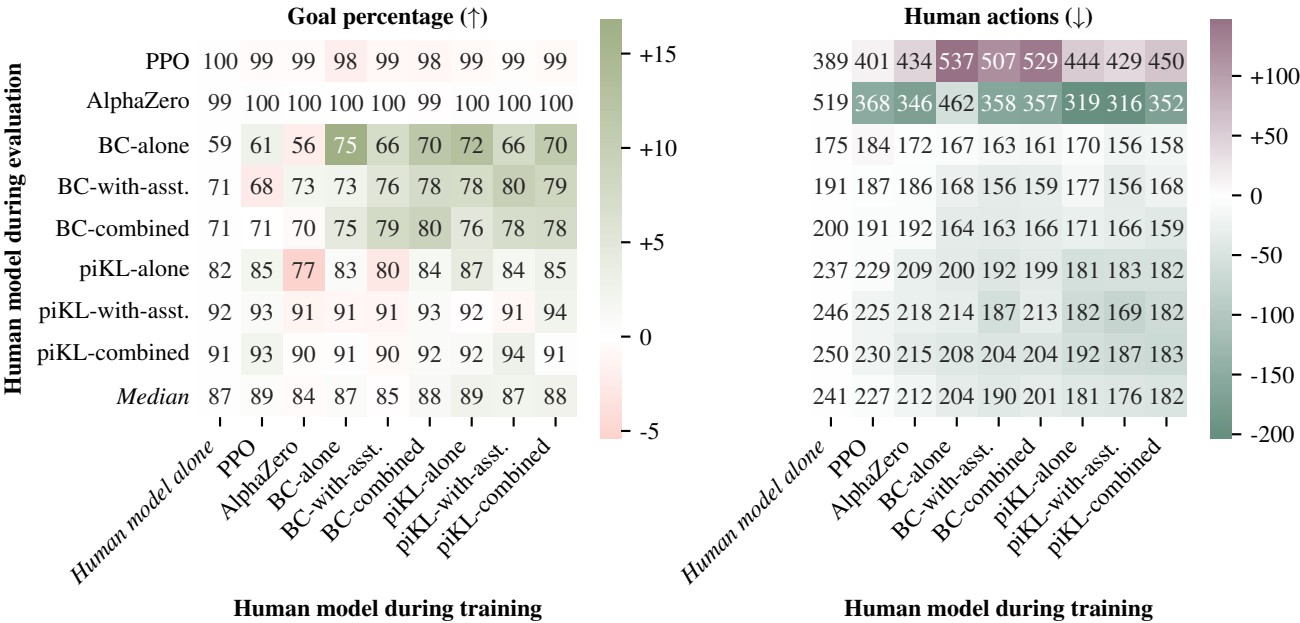

Figure 9: We train AssistanceZero assistant policies with each of our eight human models and evaluate the assistants with all human models. Here, we show the mean goal percentage achieved by each assistant-human pair as well as the mean number of place and break actions taken by the human. Colors indicate the *difference* in each metric compared to the human model building alone.

Figure 9 shows the full results of training AssistanceZero assistant policies with all of our human models and evaluating them with every other human model. We find that training the assistant with the piKL human models yields the best performance, increasing the percentage of the goal structure that is built while reducing the number of actions taken by the human model. Assistant policies trained with PPO- and AlphaZero-based human models performed the worst, demonstrating the issue with modeling humans as rational or Boltzmann-rational.

### D.1.2. BEHAVIOR CLONING ABLATIONS

We perform several ablations of our best behavior cloning model, BC-combined. The results are shown below using the same metrics as in Table 2:

| Ablation | Cross entropy | | Goal % after X min | | | |
|---|---|---|---|---|---|---|
| | Alone | w/ asst. | 3 | 5 | 10 | 20 |
| None | **1.89** | **1.99** | 9 | 17 | 41 | 71 |
| No data augmentation | 2.41 | 2.36 | 10 | 18 | 35 | 62 |
| No dropout | 2.56 | 2.44 | 8 | 14 | 30 | 49 |
| No LSTM | 2.13 | 2.12 | 12 | 21 | 43 | 70 |
| No previous action input | 2.40 | 2.36 | 12 | 22 | 44 | 71 |
| Humans subjs. | — | — | 25 | 42 | 80 | 95 |

Table 4: Ablations of key components of our BC human models. See Appendix E for the full meaning of all ablations.

The ablation study shows that data augmentation, dropout, using a recurrent network, and using the previous action as input are all important to achieving low cross entropy with BC. Furthermore, removing data augmentation or dropout also considerably lowers the performance of the BC model playing alone.

### D.1.3. PIKL ABLATIONS

As described in Appendix E.3.1, the most important hyperparameter for our piKL human models is $c_{\text{PUCT}}$, which trades off between policies that achieve higher reward versus ones that are closer to the BC model. Below, we show variations of our piKL-combined human model with various values of $c_{\text{PUCT}}$.

| | Cross entropy | | Goal % after X min | | | |
|---|---|---|---|---|---|---|
| $c_{\text{PUCT}}$ | Alone | w/ asst. | 3 | 5 | 10 | 20 |
| 10 | 2.28 | 2.61 | 39 | 60 | 82 | 92 |
| 30 | 1.98 | 2.20 | 26 | 44 | 75 | 91 |
| 50 | **1.91** | **2.08** | 21 | 36 | 65 | 88 |
| Humans subjs. | — | — | 25 | 42 | 80 | 95 |

Table 5: Ablations of the $c_{\text{PUCT}}$ parameter for the piKL-combined human model. We find that using $c_{\text{PUCT}} = 50$ achieves the lowest cross entropy, but builds houses much slower than real humans. $c_{\text{PUCT}} = 10$ builds houses *faster* than real humans and has much higher cross entropy. We decided to use $c_{\text{PUCT}} = 30$ for our main experiments because it achieves relatively low cross entropy and closely matches human performance at building houses alone.

### D.2. PPO assistant training

We conduct extensive ablation experiments to train a PPO-based assistant policy with an imitation-learning based human model, as shown in Table 6. First, we experimented with interleaving convolutional and LSTM layers or removing the LSTM layers. Next, we tried reward engineering by only providing reward based on the assistant's own actions, rather than the shared reward that also depends on the human model's actions. We also included auxiliary losses to encourage correct block placement ("block-placing loss") and predict the goal structure ("goal prediction loss"). Finally, we ablated the standard PPO entropy bonus and value function loss. The best overall policy does not include LSTM layers, utilizes reward engineering, and adds the block-placing loss in addition to the standard PPO losses. See Appendix E.3.2 for more information about PPO assistant training and the final set of hyperparameters.

| LSTM | Reward engineering | Block-placing loss | Goal prediction loss | Entropy coefficient | VF loss | Overall goal % | Human actions | Assistant goal % |
|---|---|---|---|---|---|---|---|---|
| ✓ | ✓ | ✓ | ✓ | ✓ | ✓ | $71.1 \pm 0.9$ | $201 \pm 3$ | $-1.1 \pm 1.0$ |
| ✓ | ✓ | ✓ | ✓ |  | ✓ | $71.2 \pm 1.0$ | $200 \pm 4$ | $-0.0 \pm 0.0$ |
| ✓ | ✓ | ✓ |  | ✓ | ✓ | $70.9 \pm 1.0$ | $200 \pm 4$ | $-0.0 \pm 0.1$ |
| ✓ | ✓ |  | ✓ | ✓ | ✓ | $71.0 \pm 1.0$ | $199 \pm 3$ | $0.3 \pm 0.6$ |
| ✓ | ✓ | ✓ | ✓ | ✓ |  | $70.6 \pm 1.0$ | $194 \pm 3$ | $0.8 \pm 1.0$ |
|  | ✓ | ✓ | ✓ | ✓ | ✓ | $71.5 \pm 0.9$ | $191 \pm 3$ | $2.8 \pm 1.0$ |
|  | ✓ | ✓ | ✓ |  | ✓ | $62.4 \pm 1.2$ | $206 \pm 3$ | $-14.4 \pm 1.6$ |
|  | ✓ | ✓ |  | ✓ | ✓ | $74.1 \pm 0.9$ | $191 \pm 3$ | $7.2 \pm 1.0$ |
|  | ✓ |  |  | ✓ | ✓ | $71.6 \pm 0.9$ | $201 \pm 3$ | $0.0 \pm 0.0$ |
|  | ✓ |  | ✓ | ✓ | ✓ | $70.8 \pm 0.9$ | $196 \pm 3$ | $0.6 \pm 0.9$ |
|  | ✓ | ✓ | ✓ | ✓ |  | $70.5 \pm 1.0$ | $193 \pm 3$ | $-0.0 \pm 1.3$ |
| ✓ |  | ✓ | ✓ | ✓ | ✓ | $71.1 \pm 1.0$ | $201 \pm 4$ | $-0.3 \pm 0.1$ |
| ✓ |  | ✓ | ✓ |  | ✓ | $71.4 \pm 1.0$ | $201 \pm 3$ | $-0.0 \pm 0.2$ |
| ✓ |  | ✓ |  | ✓ | ✓ | $70.5 \pm 1.0$ | $200 \pm 3$ | $-0.6 \pm 0.2$ |
| ✓ |  |  | ✓ | ✓ | ✓ | $72.9 \pm 0.9$ | $203 \pm 3$ | $0.1 \pm 0.5$ |
| ✓ |  | ✓ | ✓ | ✓ |  | $69.9 \pm 0.9$ | $207 \pm 3$ | $-4.2 \pm 0.8$ |
|  |  | ✓ | ✓ | ✓ | ✓ | $67.9 \pm 1.0$ | $195 \pm 3$ | $-3.0 \pm 0.9$ |
|  |  | ✓ | ✓ |  | ✓ | $72.0 \pm 1.0$ | $207 \pm 3$ | $-2.6 \pm 0.8$ |
|  |  | ✓ |  | ✓ | ✓ | $70.9 \pm 1.0$ | $200 \pm 3$ | $0.3 \pm 0.3$ |
|  |  |  | ✓ | ✓ | ✓ | $68.2 \pm 1.0$ | $194 \pm 3$ | $-1.0 \pm 0.9$ |
|  |  | ✓ | ✓ | ✓ |  | $71.5 \pm 0.9$ | $204 \pm 3$ | $-1.6 \pm 0.8$ |

Table 6: Full ablation results of evaluating how well PPO-based assistant policies trained with an imitation learning-based human model build goal structures not seen during training. Overall goal % is the total percentage of the goal completed; human actions refers to the number of place and break actions taken by the human model; and assistant goal % is the percentage of the goal completed by the assistant. The first six ablation columns correspond to whether LSTM layers are used; reward engineering by only providing reward for the assistant's own actions; an auxiliary loss to encourage correct block placement; a goal prediction loss; the PPO entropy bonus; and the PPO value function loss.

## D.3. AssistanceZero ablations

We present two ablations of AssistanceZero in MBAG:

| Ablation | Overall goal % | Human actions | Assistant goal % |
|---|---|---|---|
| None | $\mathbf{77.5 \pm 3.2}$ | $\mathbf{154 \pm 9}$ | $\mathbf{25.2 \pm 4.6}$ |
| No LSTM | $69.0 \pm 3.6$ | $192 \pm 11$ | $-0.6 \pm 5.2$ |
| $\lambda_{\text{prev-rew}} = 0$ | $76.8 \pm 2.6$ | $167 \pm 10$ | $18.1 \pm 5.1$ |

Table 7: Ablations of AssistanceZero.

As expected, because AssistanceZero is solving a POMDP, a recurrent policy performs much better. We also validate the inclusion of the KL penalty between the previous and current reward parameter prediction distributions (which is scaled by $\lambda_{\text{prev-rew}}$).

# E. Experiment details

Here, we provide further details about our data collection and training procedures.

## E.1. Data collection

To train the BC human models, we collect 18 episodes of 5 human subjects building goal structures. For half of the total episodes, the subject is given a goal structure and is instructed to build it quickly and efficiently without assistance. For the other half, a single experienced human Minecraft player acts as the assistant to help build the house. The human assistant is instructed to help the human subjects build their goal structures, but they are not shown the goal structure themselves. While the human agent and assistant can observe each other's actions, there is otherwise no communication between them.

Out of the five human subjects we collected data from, four were male and one was female; four had previous Minecraft experience and one did not.

## E.2. Network architecture

For both the human models and AI assistant policies, we use a convolutional neural network architecture with six residual blocks and (optionally) two LSTM blocks:

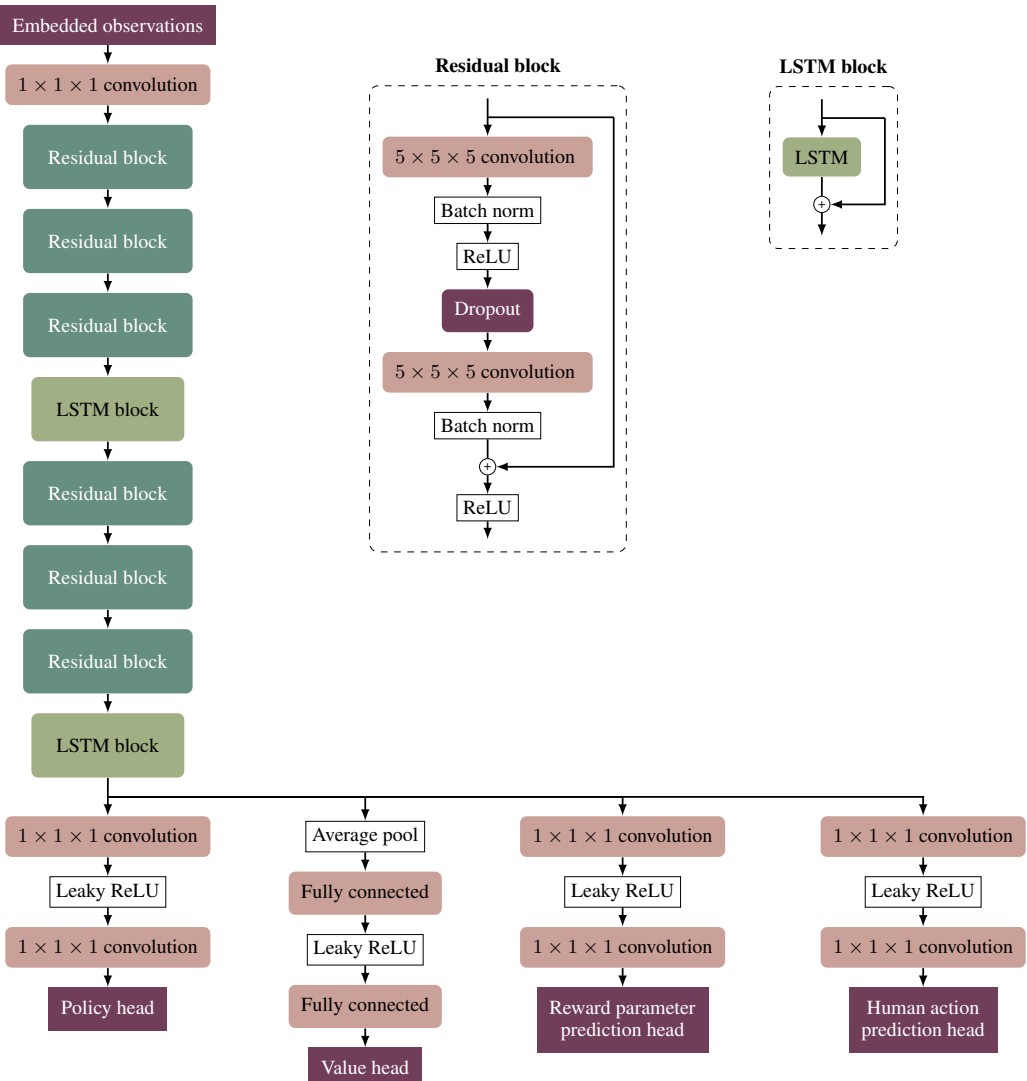

The network takes in observations as a tensor of shape $W \times H \times D \times N$ for an environment of size $W \times H \times D$, where each location includes the following features:

- an embedding representing the current block type present at that location,

- an embedding representing the goal block type at that location (if the goal is visible to the agent),

- an embedding representing which player, if any, is standing at that location,

- an embedding representing which player, if any, was the last to place or break a block at that location (this allows the agents' actions to be visible to each other),

- the counts of each type of block in each players' inventories divided by 64,

- and the current timestep divided by 1,000.

The observation embeddings are transformed via a $1 \times 1 \times 1$ convolutional layer (i.e., a fully connected layer at each spatial location) before being passed through the backbone.

The backbone consists of six or eight layers depending on whether the network is recurrent. The residual layers follow the ResNet architecture (He et al., 2016) but with 3D $5 \times 5 \times 5$ convolutions and optional dropout. An LSTM block consists of a standard LSTM layer with a skip connection, where the LSTM is applied separately at every spatial location in the input. The residual and LSTM blocks use 64 channels throughout the network.

The output of the backbone is a tensor of size $W \times H \times D \times 64$. It is passed through the four heads described in Section 4.2:

1. The *action head* consists of two $1 \times 1 \times 1$ convolutional layers with a Leaky ReLU activation function in between. The output of the action head is a $W \times H \times D \times (2B + 8)$ for a environment of size $W \times H \times D$ with $B$ block types ($B = 10$ in our experiments). The action head is passed through a softmax function to produce a distribution over actions. Each element of the output corresponds to a possible action, with some actions represented by multiple elements. Seven of the output channels correspond to the no-op and movement actions; the probabilities are summed across all spatial locations to produce a distribution over these actions. One channel corresponds to the break block action at each spatial location. $B$ channels correspond to the place block action at each spatial location, with each channel representing a different block type. Finally, the last $B$ channels correspond to the give block action at each spatial location, with each channel representing a different block type; give block actions are only valid for locations with another player that is near by. We mask invalid actions by setting their probabilities to $0$ and renormalize the distribution.

2. For the *value head*, the backbone outputs are averaged over all spatial locations to produce a single vector of dimension 64. This is then passed through two fully connected layers with a Leaky ReLU activation function in between. The output of the value head is a scalar.

3. For the *reward parameter prediction head*, the backbone outputs are passed through two $1 \times 1 \times 1$ convolutional layers with a Leaky ReLU activation function in between. The output of the goal head is a tensor of size $W \times H \times D \times B$, where $B$ is the number of block types. At each spatial location a softmax is applied; this produces a predicted distribution over the block types in the goal structure at that location.

4. The *human action prediction head* has an identical architecture to the policy head. The output of the human action prediction head is a distribution over actions that the human is likely to take, with the outputs interpreted the same way as the policy head.

### E.3. Training details

We implement all RL and imitation learning algorithms in RLlib (Liang et al., 2018) and PyTorch (Paszke et al., 2019). During RL training, we randomize the starting location of the human policy to improve generalization. Since some RL algorithms sample experience in fragments shorter than a full episode, we also randomize the length of the first episode in the environment. This avoids a situation where in one iteration of PPO all fragments are from the beginning of episodes and in the next they are all from the end.

### E.3.1. IMITATION LEARNING

We use behavior cloning for our BC human models as well as the pretraining and SFT assistants.

**Data augmentation** We use data augmentation during behavior cloning for some experiments. The data augmentation consists of choosing a random permutation of block types for each state and applying it to the current blocks in the world, the block types in the goal structure, the players' inventories, and any place or give actions. We found that data augmentation helped in some cases; see the BC ablations in Appendix D.1.2 and the details of the SFT assistant training in Appendix E.3.1.

**Behavior cloning human models** As described in the main text, we train human models with behavior cloning on three datasets: 9 episodes of humans playing alone, 9 episodes of humans playing with an assistant, and the full dataset of 18 episodes (see Appendix E.1). We use the network architecture described in Appendix E.2 for our BC models, but with an additional input of the previous action taken by the human model. We found that this substantially improved human action prediction (see ablations in Appendix D.1.2). We use the following hyperparameters:

| Hyperparameter | Value | | |
|---|---|---|---|
| | BC-alone | BC-with-assistant | BC-combined |
| Epochs | 30 | 80 | 40 |
| Data augmentation | | yes | |
| LSTM | | yes | |
| Dropout | | 0.7 | |
| SGD batch size | | 128 | |
| Optimizer | | Adam | |
| Learning rate | $10^{-3}$ decayed linearly to $10^{-4}$ over first half of training | | |

Table 8: Hyperparameters for BC human models.

The only difference between the models trained on different splits was the number of epochs. See Appendix D.1.2 for ablations of these hyperparameters.

**piKL human models** piKL (Jacob et al., 2022) is a human model that combines a BC-trained policy with MCTS. In particular, piKL selects actions by running MCTS with the prior policy given by the BC network's output. Grill et al. (2020) show that this is approximately equivalent to solving a regularized optimization problem that finds the policy which maximizes reward minus a KL constraint to the BC policy.

We carefully tune the parameter $c_{\text{PUCT}}$ in MCTS which effectively interpolates between purely maximizing reward and purely following the BC policy (see Appendix D.1.3). We find a value of 30 balances prediction error and performance.

A drawback of using piKL as a human model is that it does assign positive probability to all actions, only those visited by MCTS. This means that the cross entropy of piKL on human data is infinite if there is a single action taken by the human that MCTS does not visit. To fix this, we define a distribution with full support over all actions based on the asymptotic approximation given in Grill et al. (2020) of the policy MCTS would reach after infinitely many simulations. We use this full-support policy for calculating the cross entropy of piKL, for evaluating piKL human models in MBAG, and while training assistants with piKL human models.

We do not use a value function for piKL, although Jacob et al. (2022) experiment with this. When running piKL in MBAG with another agent, we plan in MCTS as though the other agent only takes no-ops.

**Pretrained assistant** To train the pretrained assistant described in Section 5, we sample 10,000 episodes from the BC-combined model. We remove information about the goal structures, segment each episode into fragments of length 64, and train a recurrent policy with the following hyperparameters:

| Hyperparameter | Value |
|---|---|
| SGD batch size | 256 |
| Total training batches | 96,000 |
| Data augmentation | no |
| LSTM | yes |
| Dropout | 0.5 |
| Optimizer | Adam |
| Learning rate | $10^{-3}$ |

Table 9: Hyperparameters for the pretrained assistant.

When evaluating the policy, we sample from it with temperature 0.3. That is, we scale the output logits by $1/0.3$ before applying softmax to obtain action probabilities.

**SFT assistant**   The SFT assistant is fine-tuned from the pretrained assistant using BC on expert human assistant data from our data collection sessions (Appendix E.1). We carefully tuned the hyperparameters of the SFT assistant using grid search over 540 parameter combinations. We trained an SFT assistant with each set of parameters and then evaluated it with the BC-combined human model for 100 episodes. We ranked the parameter combinations based on the percentage of the goal built on the assistant. Then, we re-evaluated the top 20 hyperparameter combinations for 1,000 episodes to reduce variance. We selected our final hyperparameter settings based on the best-performing assistant from these evaluations according to goal percentage built by the assistant.

The table below shows the final parameters as well as those considered in the grid search:

| Hyperparameter | Value | Values considered in grid search |
|---|---|---|
| Initialization | Pretrained assistant w/o action head | { Random, pretrained assistant w/ or w/o action head } |
| Training epochs | 100 | $\{10, 20, 30, 50, 100\}$ |
| Data augmentation | yes | $\{yes, no\}$ |
| LSTM | yes | — |
| Dropout | 0 | $\{0, 0.5\}$ |
| Optimizer | Adam | — |
| SGD batch size | 256 | — |
| Learning rate | $10^{-4}$ | $\{10^{-3}, 3 \times 10^{-4}, 10^{-4}\}$ |
| Sampling temperature | 0.3 | $\{1, 0.5, 0.5\}$ |

Table 10: Hyperparameters for the SFT assistant. We tune the hyperparameters via grid search over the values in the right column, if given. We consider initialization of the policy network from either random weights or from the weights of the pretrained assistant. Initialization w/o the action head means we initialize all weights from the pretrained assistant except for those in the action head.

E.3.2. REINFORCEMENT LEARNING

**PPO human model (single-agent)**   We use the following hyperparameters to train the PPO human model, which we trained to build houses alone:

| Hyperparameter | Value |
|---|---|
| Training iterations | 100 |
| Rollout length | 500 |
| Number of environments | 640 |
| SGD batch size | 512 |
| SGD epochs per iteration | 3 |
| Optimizer | Adam |
| Learning rate | $3 \times 10^{-4}$ |
| Discount factor ($\gamma$) | 0.95 |
| GAE coefficient ($\lambda$) | 0.95 |
| Entropy coefficient | 0.03 |
| Clipping parameter | 0.2 |
| Gradient clipping | 10 |
| LSTM | No |
| Dropout | 0 |
| KL target | 0.01 |
| Initial KL coeff. | 0.2 |
| Value function loss coeff. | 0.01 |

Table 11: Hyperparameters for PPO human model training.

**PPO assistant**    To effectively train an assistant with PPO, we modified the reward function and added an auxiliary loss term. For the former, we only give reward that is directly attributable to the place/break actions of the assistant and disregard any place/break actions taken by the human. This means that PPO's goal is not actually aligned with the assistance game objective. However, without this modification, we found that the PPO assistant did not make meaningful contributions to building the goal structure—it either took no-op and movement actions or repeatedly placed and broke the same block.

For the auxiliary loss, which we call the "block-placing loss," we use the cross-entropy between the block type placed by the assistant and the goal block type at that location, if there is one. This loss provides some training signal when the assistant places a block in a location that is part of the goal structure, even if the block type is incorrect. Without this loss, placing an incorrect block type would simply result in a reward of 0, making it more challenging for the assistant to learn to place blocks at all. We linearly decay this loss coefficient from 1 to 0 over the first $2 \times 10^6$ timesteps.

We also experimented with adding a second auxiliary loss term to predict the goal structure. This involved adding a goal prediction head similar to that used in AssistanceZero and training with the same loss function. However, we did not find that this loss produced the best PPO assistant.

Finally, we observed that removing the LSTM blocks from the baseline network architecture described in Appendix E.2 improved the assistant's performance.

All the hyperparameters for the PPO assistant are shown in Table 12. See Appendix D.2 for a full list of ablation experiments and results.

| Hyperparameter | Assistant |
|---|---|
| Training iterations | 300 |
| Rollout length | 64 |
| Number of environments | 256 |
| SGD minibatch size | 256 |
| SGD epochs per iteration | 3 |
| Optimizer | Adam |
| Learning rate | $3 \times 10^{-4}$ |
| Discount factor ($\gamma$) | 0.95 |
| GAE coefficient ($\lambda$) | 0.95 |
| Entropy coefficient (horizon) | $3 \rightarrow 0.01 \ (2 \times 10^6)$ |
| Clipping parameter | 0.2 |
| Grad clip norm threshold | 10 |
| Recurrent network (LSTM) | No |
| KL target | 10 |
| KL coeff. | 0.2 |
| Value function coeff. | 0.01 |
| Goal loss coeff. | 0 |
| Place block loss coeff. (horizon) | $1 \rightarrow 0 \ (2 \times 10^6)$ |

Table 12: Hyperparameters for PPO assistant training.

**MCTS**  Actions in MBAG consist of a high-level action type (no-op, break block, place block, move up, etc.) and parameters for the location (used by break/place) and block type (used by place). Because of this structure, we found it helpful to separate the action selection step of MCTS into two stages, which we refer to as bi-level action selection. First, MCTS chooses the high-level action type by using aggregated prior policy probabilities, Q-values, and visit counts that are summed over all actions with that action type. Then, if the action type requires additional parameters (i.e., place and break actions), we repeat the action selection process among all actions of that type.

Similarly to AlphaZero, we add Dirichlet noise to the action selection step. We use separate noise levels for the two stages—0.25 for the first action type stage, and 10 divided by the number of valid actions for the second stage.

**AlphaZero human model (single-agent)**  We use the following hyperparameters to train the AlphaZero human model to build houses alone:

| Hyperparameter | Value |
|---|---|
| Training iterations | 125 |
| Rollout length per iteration per environment | 64 |
| Number of environments | 256 |
| Replay buffer size | 65,536 |
| Timesteps sampled from replay buffer per iteration | 65,536 |
| SGD batch size | 256 |
| SGD epochs per iteration | 1 |
| Optimizer | Adam |
| Learning rate | $10^{-3}$ |
| Discount factor ($\gamma$) | 0.95 |
| Gradient clipping | 10 |
| LSTM | no |
| Dropout | 0 |
| Value function loss coeff. | 0.01 |
| No-op reward | -0.2 |
| Number of MCTS simulations | 100 |
| Inverse temperature for MCTS | 1.5 |
| $c_{\text{PUCT}}$ | 1 |

Table 13: AlphaZero hyperparameters for the human model (single-agent) and assistant training.

We used two additional tricks to improve single-agent AlphaZero training. First, we terminate episodes if a new minimum goal distance is not achieved for 100 timesteps. Second, we add a penalty to the reward function of $-0.2$ for no-op actions to encourage the policy to act and explore.

**AssistanceZero assistant**   We use the following hyperparameters for training assistants with AssistanceZero:

| Hyperparameter | Value |
|---|---|
| Training iterations | 500 |
| Rollout length per iteration per environment | 64 |
| Number of environments | 256 |
| Replay buffer size | 262,144 |
| Timesteps sampled from replay buffer per iteration | 65,536 |
| SGD batch size | 256 |
| SGD epochs per iteration | 1 |
| Optimizer | Adam |
| Learning rate | $10^{-3}$ |
| Discount factor ($\gamma$) | 0.95 |
| Gradient clipping | 10 |
| LSTM | yes |
| Dropout | 0 |
| Number of MCTS simulations | 100 |
| Inverse temperature for MCTS | 1.5 |
| $c_{\text{PUCT}}$ | 1 |
| $\lambda_{\text{policy}}$ | 1 |
| $\lambda_{\text{value}}$ | 0.01 |
| $\lambda_{\text{reward}}$ | 3 |
| $\lambda_{\text{prev-rew}}$ | linear increase from 0 to 30 over training |
| $\lambda_{\text{action}}$ | 1 |

Table 14: AssistanceZero hyperparameters for MBAG.

### E.4. Evaluation

When evaluating AssistanceZero assistants, we use only 20 simulations of MCTS, which is roughly the number that can run in real-time with Minecraft on an NVIDIA GeForce 1080 Ti GPU. All evaluations use randomly sampled houses from the test set $\mathcal{D}_{\text{test}}$, while all training uses houses from the train set $\mathcal{D}_{\text{train}}$; thus, we always test human models and assistants on unseen goal structures.

