# OpenReview forum: "AssistanceZero: Scalably Solving Assistance Games"
_ICML.cc/2025/Conference — ICML 2025 poster_

### Official Review · Reviewer_mxi5 · 2025-03-09

**Overall Recommendation:** 3

**Summary:**

The paper proposes AssistantZero, a method to learn cooperative assistants where the reward function of the human player is unknown. The paper presents a new environment, MBAG-- a 3D grid with many possible reward functions and a variety of ways to help the human in the loop. In contrast to using a PPO, AssistantZero is similar to AlphaZero and uses MCTS equipped with a reward model and network that predicts human actions. Experiments on MBAG show that AssistantZero outperforms PPO.

**Claims And Evidence:**

- The introduction mentions that existing methods for assistance methods do not scale well and are used in toy settings. However, this paper presents results on a 3D grid world.

- It is hard to articulate the second challenge around lines 104 that human model is needed to predict human's response and is a challenges for methods using assistance games. The paper presents a result with a trained human model and it is not clear how to get around this?

**Essential References Not Discussed:**

NA

**Experimental Designs Or Analyses:**

The details of the baselines are not clear, see Q4 for more details.

**Methods And Evaluation Criteria:**

This paper proposes a new environment MBAG to test if the assistant agent can help the human model to solve the task efficiently. For toy setting, the environment is fine. However, there are no environments to test if the method can scale.

**Other Comments Or Suggestions:**

NA

**Other Strengths And Weaknesses:**

## Strengths
- The idea of learning cooperative agents where the reward function is not known to the assistant is interesting and challenging.
- The proposed method AssistantZero shows promising results and leads to lower overall steps needed to solve tasks in environment.

## Weaknesses
- The introduction mentions that existing methods for assistance methods do not scale well. However, this paper presents results on a 3D grid world.
- The method relies on predicting the parameters of the reward model which might not scale to larger models.
- Some details are hard to follow. For example, the details of PPO training and why does the assistant need a non-Markovian policy for environments like MBAG.

**Questions For Authors:**

1. For the reward function, AssistantZero predicts the parameters of the network. How will this scale when the reward function is complex or to a setting like RLHF where the reward model is built on top of an LLM?
2. Although the paper is trying to solve assistant games, the policy of human is kept constant which is often not true in many scenarios where both human and assistant are trying to recover the reward and goal in the environment. How will this method scale to such scenarios?
3. How would exploration methods help in setting where the reward function is sparse?
4. What was the loss function used to train the PPO baseline and how does it differ with Eq 1?
5. How expensive is MCTS compared to PPO at test-time in terms of computational requirements?

**Relation To Broader Scientific Literature:**

This paper presents a method to learn assistant agents that can aid humans in solving tasks more efficiently. The proposed method uses a MCTS for planning at inference. Learning a best-response assistant player makes sense, but the paper does not discuss the setting where the human player (ideally learned with a no-regret strategy) does not have access to the reward function and is learning the task with the assistant.

**Theoretical Claims:**

There are no theoretical claims in the paper.

---

> ### Author Rebuttal · Authors · 2025-04-01
>
> We thank the reviewer for their thorough review and helpful feedback. Below are responses to individual points and questions:
>
> * **Complexity of environment:** While the reviewer refers to MBAG as a toy setting, we argue it is substantially more complex than prior environments in assistance games, which used 5x5 or 9x10 grid worlds with less than a dozen possible goals like collecting lemons or gemstones [1, 2]. MBAG instead involves building houses in Minecraft, a more realistic task with over $10^{400}$ possible goals. That PPO struggles even with reward engineering and auxiliary losses underscores the environment's difficulty.
>
>   Overcooked [3] and Hanabi [4], despite being simpler than MBAG, have enabled significant research on collaborative AI—further suggesting MBAG is a meaningful benchmark.
> * **“It is hard to articulate the second challenge around lines 104 that human model is needed...”** This is a challenge articulated in past work, e.g., Fisac et al. [5] write that “more realistic human models [are] therefore crucial in using the CIRL formulation to solve real-world value-alignment problems” (CIRL = assistance games). While past papers have used human models based on RL or planning [1, 2], we are the first to leverage an imitation learning-based human model for assistance games. We will update the introduction to clarify this.
> * **“AssistantZero predicts the parameters of the [reward] network. How will this scale?”** The reward function $R(s\_t, a\_t^\\textbf{H}, a\_t^\\textbf{R}; \\theta)$ is parameterized by $\\theta$, but is not generally a *neural network*. The parameters $\\theta$ can be any information that encodes the task/goal/preferences of the human. In our case, $\\theta$ consists of the goal structure, i.e., the configuration of blocks that the human is attempting to assemble. In the case of an LLM assistant for coding, $\\theta$ might consist of a set of test cases that the human would like to satisfy by implementing some function. We will modify Section 2 to make this more clear.
> * **"Some details are hard to follow. For example, the details of PPO training…”** We used the standard PPO loss, which consists of the clipped surrogate objective, the value function loss, and the entropy bonus. We also experimented with modifications to try to boost PPO's performance, as described in Section 4.1 and Appendix D.2. Equation (1), the AssistanceZero loss function, has the policy loss, value loss, reward loss, previous reward KL loss, and the human action prediction loss. Only the value loss is identical between PPO and AssistanceZero. We did try adding the goal prediction loss to PPO but found it did not improve performance (see Appendix D.2).
> * “**Why does the assistant need a non-Markovian policy?”** Past work on assistance games [6] has shown that finding a best response to a fixed human policy is equivalent to solving a single-agent POMDP. Solving a POMDP requires a non-Markovian policy and thus we use a recurrent policy for the assistant.
> * **“The policy of human is kept constant which is often not true...”** We agree that this is not a completely realistic assumption. If the human is also learning about their own reward/goal, then their behavior may be non-Markovian; this setting of “partially observable assistance games” is studied in a recent paper [7]. Humans could also adapt to a particular assistant policy, making their policy non-constant. However, we find that our assistant still works well with real humans, indicating that our assumption of a fixed human policy is reasonable in practice.
> * **“How would exploration methods help in setting where the reward function is sparse?”** MCTS already inherently provides exploration as part of AssistanceZero. Future work could investigate reward shaping, pretraining, and intrinsic motivation to further improve performance in sparse-reward settings.
> * **“How expensive is MCTS compared to PPO at test-time?”** We use 20 simulations of MCTS at test time, requiring 20 forward passes. Despite this, it runs in real time on a 1080 Ti GPU. Notably, AssistanceZero without test-time MCTS matches performance, using the same compute as PPO:
>
> Assistant|Forward passes per action|Overall goal %|Human actions|Assistant goal %
> -|-|-|-|-
> PPO w/o LSTM + rew. engineering and aux. loss|1|74.1 ± 0.9 |191 ± 3|7.2 ± 1.0
> AssistanceZero w/ test-time MCTS|20|79.8 ± 0.9|**158 ± 3**|27.0 ± 1.5
> AssistanceZero w/out test-time MCTS|1|**80.2 ± 0.9**|**158 ± 3**|**27.3 ± 1.3**
>
> [1] Woodward et al. Learning to Interactively Learn and Assist.
>
> [2] Zhi-Xuan et al. Pragmatic Instruction Following and Goal Assistance.
>
> [3] Carroll et al. On the Utility of Learning about Humans for Human-AI Coordination.
>
> [4] Bard et al. The Hanabi challenge: A new frontier for AI research.
>
> [5] Fisac et al. Pragmatic-Pedagogic Value Alignment.
>
> [6] Shah et al. Benefits of Assistance Over Reward Learning.
>
> [7] Emmons et al. Observation Interference in Partially Observable Assistance Games.

---

### Official Review · Reviewer_Gkck · 2025-03-10

**Overall Recommendation:** 4

**Summary:**

The paper proposes using assistance games instead of RLHF to train AI assistants. The advantages of assistive games are that 1) they make AI and humans collaborate to accomplish tasks instead of AI just trying to get good ratings from humans (by potentially fooling humans); 2) AI reasons with uncertainty about human's true intention. The paper introduces a benchmark assistive game named MBAG, which is based on MineCraft and is complex enough while reasonable to work with for humans and academic labs. The authors introduce AssistanceZero, the first algorithm that can perform well in complex assistive games and outperform other assistance paradigms. This shows possibility to develop AI assistants via assistive games, which are previously seen as intractable and hard to be scaled to train AI assistants. The paper also explores approaches to model human behaviors in MBAG game to use as surrogate human models in AssistanceZero training. Overall, the paper is well-written, makes strong contributions, and provides meaningful results for several areas of audience. Therefore, I lean towards accepting the paper.

## Update after rebuttal
Author's responses clearly answer my questions and the additional explanation and discussion will greatly help show the paper's value beyond controlled settings. As such, I keep my score.

**Claims And Evidence:**

Yes

**Essential References Not Discussed:**

Not that I am aware of

**Ethical Review Concerns:**

It seems unclear whether an IRB was obtained for the human study.

**Ethical Review Flag:**

Flag this paper for an ethics review.

**Ethics Expertise Needed:**

["Responsible Research Practice (e.g., IRB, documentation, research ethics, participant consent)"]

**Experimental Designs Or Analyses:**

Yes. The experiment designs for evaluating AssistanceZero, human modeling, and final human user study are sound and valid.

**Methods And Evaluation Criteria:**

Yes. MCTS makes sense as a method to solve assistive games and the authors augmented it with a model to predict the latent objective and human-teammate actions. The performance metrics of overall task performance and human assistance provided make sense.

**Other Comments Or Suggestions:**

There is a "1" on line 129 after word "action".

**Other Strengths And Weaknesses:**

Despite the strenths of the paper, I do have a few suggestions for further improving the paper.
1. In the intro, it seems the authors indicate the key difference between RLHF and Assistance game is whether the AI collaborates with the human on the task (in assistive games, the AI does). However, I think this teaming or collaboration may not be the most important key - for example, in a chat system the robot still only suggests ways to solve the human's problem, e.g., to fix a coding bug. There is no "action" the agent can actually take. I think the key is that the agent is also getting a ground-truth reward signal instead of just a human's thumbs-up/down. Even in chat system, the agent should aim to solve the human's problem, instead of fooling humans into thinking the problem is solved while it is actually not. In MBAG, this is also the key - the agent is getting the environment's feedback about positive or negative reward regarding the task progress (whether a block has been placed/destroyed correctly or not).
2. The training of both the human model and the assistive model seem to assume that humans are homogeneous - one model could represent human and one model could represent how to assist human. Is this a limiting assumption that future work should seek to relax? Does the authors observe heterogeneity in human data and preferences during experiment?

**Questions For Authors:**

1. The paper assumes access to $\hat{p}_t{\theta}$ in Line 294. How can this be available for general human tasks? Is this a restricting assumption? If so, making the limitation clear would be great.
2. More generally, is assuming the reward structure as known a restricting assumption that blocks application to general AI training? It seems unclear how human reward can be structured with a reasonable amount of parameters. What is the reward structure used in MBAG and how large is the $\theta$ space in the experiment?
3. What if the reward structure prior is incomplete or even wrong?

**Relation To Broader Scientific Literature:**

The paper proposes to replace RLHF with assistive games due to the advantages mentioned in the paper summary. The assistive games were previously seen as intractable and the paper proposes a model-free RL with MCTS to potentially mitigate the issue.

**Theoretical Claims:**

There is no theoretical claim in the work.

---

> ### Author Rebuttal · Authors · 2025-04-01
>
> We thank the reviewer for their thorough review and helpful suggestions. Below are responses to individual questions.
>
> ## Responses to questions
>
> **1\. The paper assumes access to $\hat{p}_t(\theta)$ in Line 294\. How can this be available for general human tasks?**
> We believe the reviewer is asking about how $\theta$ (the true reward parameter) is available during training. $\theta$ is not visible to the assistant policy when it is actually acting in the environment; the assistant infers a belief distribution over $\theta$ using the reward parameter prediction head. However, once an episode completes, the $\theta$ value that was used by the human model during that episode *is* available for updating the network to enable better assistance in the future, and thus we can use it as part of the loss function for AssistanceZero.
>
>
> **2\. More generally, is assuming the reward structure as known a restricting assumption that blocks application to general AI training? … What is the reward structure used in MBAG and how large is the $\theta$ space in the experiment?**
>
> In our MBAG environment, the goal parameters $\theta$ consist of the specific arrangement of blocks in the goal house that the human is trying to build. This space contains over $10^{400}$ possible goal structures, since each block in the 3D grid that makes up the Minecraft world can be 10 different types. In a more complex setting, like a coding assistant, the goal parameters could instead consist of a set of unit tests that the user wants to satisfy by writing a function. One could collect a large dataset of function specifications along with unit tests and define the reward function based on how many unit tests pass at the end of the human-assistant interaction. In an even more general setting, an LLM judge could be used to give rewards based on detailed task descriptions that are used as the reward parameters. This would be distinct from RLAIF because a separate human model is used to interact with the assistant while a powerful LLM judge evaluates whether the latent task is actually accomplished. While these generalizations of our work still require overcoming additional challenges, we believe that our approach can scale to complex assistance settings.
>
> **3\. What if the reward structure prior is incomplete or even wrong?**
>
> We note that our reward structure in MBAG is likely incomplete. It only describes the goal house and doesn’t represent other preferences the human may have for the assistant’s behavior, such as not getting in their way or obstructing their view. Despite this, our assistant performs well with real humans in the user study.
>
> Nevertheless, there are a few ways in which we can handle an incomplete/incorrect reward structure prior. One way is to detect if the human’s behavior is too unlikely under any reward function we can represent, and perform safe fallback actions. Another approach is to combine assistance games with RLHF by including an RLHF loss term that represents human preferences not described in the structured reward model. This is a good direction for future work, which we will mention in the conclusion.
>
> ## Key benefits of assistance games
>
> **“the key is that the agent is also getting a ground-truth reward signal instead of just a human's thumbs-up/down.”**
>
> We agree; receiving reward from the latent “true” reward function rather than (potentially biased) feedback from the human is a key ingredient in assistance games. We appreciate the reviewer’s point that the assistance game approach would encourage the AI to be more proactive in order to maximize the shared reward rather than simply react to and please the human. We'll emphasize this important distinction more clearly in the paper.
>
> ## Human heterogeneity
>
> **“The training of both the human model and the assistive model seem to assume that humans are homogeneous… Is this a limiting assumption that future work should seek to relax? Does the authors observe heterogeneity in human data and preferences during experiment?”**
>
> While we use a single human model, it can still represent heterogeneous behavior. Theoretically, since we use an LSTM-based human model, it can represent a mixture of human policies. In fact, we expect this to be the case since we observe significant heterogeneity in the human training data ( and the user study). Participants had very different levels of experience with playing Minecraft and computer/video games, so their building speed and play styles varied greatly.

---

> > ### Comment · Reviewer_Gkck · 2025-04-03
> >
> > Thanks for posting author response!
> >
> > Q1: to clarify, I was asking about how we know the ground-truth prior distribution about $\theta$, which is $\hat{p}_t(\theta)$. It is understandable that in the MineCraft study this is designed by the experimenter and is known. However, how do we know about this for real-world applications?
> >
> > Q2: I understand in MBAG, the space of $\theta$ contains large number of possible goal structures due to the composition, but I was mostly asking how you represent that with $\theta$. I am guessing that it is a 400-length vector with each item being 0-9 as the 10 different types? This is reasonable to do in MBAG and make it feasible to calculate the likelihood and KL-divergence to the prior in Equation 1. However, how does this extend to less-structured real-world problems, e.g., how do you calculate the likelihood in your LLM-judge example?
> >
> > Human heterogeneity: very interesting extra information you mentioned! More in-depth analysis on how assistance game training helps different kinds of people would be a great add to the work.

---

> > > ### Author Response · Authors · 2025-04-03
> > >
> > > **Q1: I was asking about how we know the ground-truth prior distribution about $\theta$...**
> > >
> > > Thank you for clarifying your question and sorry for misinterpreting it\! We believe you are asking about $p(\theta)$, the prior distribution from which reward parameters are sampled at the beginning of an episode (defined on line 153). In our case, since each $\theta$ value represents a goal structure, we implicitly define $p(\theta)$ by a fixed dataset over goals, which consists of around 1,000 Minecraft houses. We split this into a training set and a test set; we use the training set for collecting human data and training all our assistants, and then evaluate whether they can assist with building previously unseen houses using the test set. This ensures that the assistants are not simply memorizing a fixed dataset, but can actually generalize to the large space of possible goals.
> > >
> > > In real-world applications, one could collect a similar dataset of “tasks” or “goals”. For example, for coding, one could curate a large set of coding tasks by using already-written code with unit tests in public repositories; for code without unit tests one could write tests with an LLM. This dataset would likely be representative of new coding tasks since it would cover a wide range of domains. For more general tasks, one could use a dataset of human-LLM conversations to extract task descriptions requested by humans; an LLM judge would assess whether the initial task is completed successfully. See below for more details on how this could be implemented in practice.
> > >
> > > **Q2: in MBAG, the space of $\theta$ contains large number of possible goal structures... However, how does this extend to less-structured real-world problems, e.g., how do you calculate the likelihood in your LLM-judge example?**
> > >
> > > In MBAG $\theta$ is represented as an 1100-length vector (11x10x10 gridword) with each item being 0-9. There is always a margin of air and ground around the house, which is why the effective goal space contains around $10^{400}$ structures and not $10^{1100}$.
> > >
> > > In less structured settings, it might not be feasible to directly predict a distribution over $\theta$, as you pointed out. However, the goal prediction head in AssistanceZero is only used to predict the intermediate rewards during MCTS. Thus, one could replace the goal prediction head with a head that directly predicts the expected reward $\mathbb{E}\[R(s, a) \mid h\]$ for an action given the history observed by the assistant, marginalizing over the reward parameters $\theta$. This head could be trained via MSE loss on observed rewards from rollouts, and would avoid the need to directly calculate a probability distribution over $\theta$.
> > >
> > > Even in more complex settings, it may still be possible to predict $\theta$ directly. In the LLM code assistant setting, $\theta$ could be a set of unit tests. One could add a $\theta$ prediction head to an LLM backbone to predict the unit tests directly since they are represented as code. The likelihood term would be the likelihood of the unit tests under the $\theta$ prediction head. In the more general LLM judge setting, $\theta$ could be a task description (e.g., “summarize this document for a lay audience”), which the $\theta$ prediction head could predict directly as well.
> > >
> > > Note that the KL divergence term isn’t strictly necessary to apply the assistance game framework. We used it for the MBAG assistant policy because it helped in practice, but it could be omitted in other domains. Nevertheless, for autoregressive LLMs, the KL divergence term can be approximated using Monte Carlo sampling. For example, this method is used in RLHF for approximating the KL divergence between the trained policy and the reference policy.
> > >
> > > **Human heterogeneity: More in-depth analysis on how assistance game training helps different kinds of people...**
> > >
> > > We agree that this is an interesting direction to look into\! We did some additional analysis of our human study data where we split the participants into two groups: experienced (≥100 hours playing Minecraft, 9 participants) and inexperienced (\<100 hours, 7 participants). The table below shows the average Likert ratings of helpfulness by both groups for each of the three assistants we tested, with 90% confidence intervals (since the groups are smaller the confidence intervals are quite wide).
> > >
> > > |Assistant|Experienced rating | Inexperienced rating |
> > > |-|-|-|
> > > | SFT | 1.6 ± 0.5 | 1.9 ± 0.3 |
> > > | AssistanceZero | 3.3 ± 0.4 | 2.9 ± 0.7 |
> > > | Human | 4.6 ± 0.3 | 3.3 ± 0.9 |
> > >
> > > Interestingly, inexperienced players tended to rate both the human and AssistanceZero assistants lower, and they rated them more closely together (although the confidence intervals are quite wide, indicating high variance). This could mean that inexperienced players have more difficulty judging how good an assistant is. Or, it could be that it is easier to assist more experienced players. It would be interesting to explore this more in future work.

---

### Official Review · Reviewer_YoBP · 2025-03-12

**Overall Recommendation:** 3

**Summary:**

This paper applies assistance games—where the assistant must infer the user’s hidden goal—to a challenging Minecraft building task with over $10^{400}$ possible goals. Their new AssistanceZero algorithm extends AlphaZero to partial observability by predicting both the human’s actions and the reward parameters (i.e., the hidden goal), then using MCTS to plan under that uncertainty. Compared to PPO and supervised fine‐tuning (as in RLHF), AssistanceZero yields assistants that better reduce the user’s workload and more accurately help complete tasks, validated by both simulation and a real user study.

**Claims And Evidence:**

Overall, most of the paper’s claims about their new algorithm (AssistanceZero) and its superior performance are backed by both simulation results and a user study in Minecraft. However, there are two concerns: (1) The user study is fairly small (only 16 participants), which limits the representativeness of the findings. (2) The paper assumes human behavior is purely Markovian with respect to $(s,\theta)$, whereas in reality people often rely on history (especially prior mistakes), which the model does not fully capture.

**Essential References Not Discussed:**

The related works are properly cited.

**Experimental Designs Or Analyses:**

The experiments are well-structured—comprising both simulation and a user study—but the user study's small sample size (16 participants) limits the generalizability of the findings. Additionally, while the paper critiques RLHF and proposes AssistanceZero as an alternative, it only compares against a pretraining plus SFT baseline rather than a full RLHF system. This leaves open the question of how AssistanceZero would perform compared to a complete RLHF approach.

**Methods And Evaluation Criteria:**

Yes. The paper’s methods—particularly the AssistanceZero algorithm (a partial‐observability extension of AlphaZero)—and evaluation criteria are well aligned with the assistance‐game setting they propose. They also introduce a Minecraft Building Assistance Game (MBAG) benchmark, which is large‐scale (over 10^{400} possible goals) and challenging enough to meaningfully test algorithms in an interactive collaboration scenario.

**Other Comments Or Suggestions:**

I don't have other comments.

**Other Strengths And Weaknesses:**

Strengths: The paper proposes a novel approach—AssistanceZero—which creatively adapts AlphaZero to a partially observable assistance game setting, addressing key limitations of RLHF. The introduction of the challenging Minecraft Building Assistance Game (MBAG) benchmark is also a strong contribution, as it pushes the problem to a more realistic and complex domain. Additionally, the experimental evaluation, combining simulation results with a human user study, provides compelling evidence for the method’s potential.
Weakness: This will be discussed in other sections.

**Questions For Authors:**

- Have you considered evaluating AssistanceZero against a full RLHF system? What challenges or limitations do you foresee in such a comparison?
- Your model assumes that human behavior is Markovian (dependent solely on the current state and reward parameters). How might your approach perform if humans exhibit non-Markovian behavior—such as incorporating historical context or prior mistakes—and have you explored this possibility?
- In future work, the authors briefly discuss LLM post-training using assistant games. However, I have several questions for discussion (not affecting my score):
    + In the real world, humans do not have full observation of the state, which is why they might ask an LLM assistant for help. This paper, however, assumes that the human has full observation. How can the ideas and framework be extended to training LLM assistants under such conditions?
    + If a human wants to complete a task and asks an LLM for help, the most efficient approach would be to communicate their intentions to the AI at the beginning (via language) so that the AI can have full observation of the goal. Given this, it is unclear how an assistant game framework would be beneficial in such cases.

**Relation To Broader Scientific Literature:**

The paper builds on several strands of prior work. First, it extends the concept of assistance games—also known as cooperative inverse reinforcement learning or hidden-goal MDPs—which were introduced by Fern et al. (2014) and further formalized by Hadfield-Menell et al. (2016). These works established that, under full observability of the state (and the human knowing the reward parameters), an optimal human policy can be defined as depending only on $(s,\theta)$.
Second, the paper leverages ideas from model-based reinforcement learning, particularly those underlying AlphaZero (Silver et al., 2017) and MuZero (Schrittwieser et al., 2020). By extending these algorithms to a partially observable setting, the authors create AssistanceZero, which uses Monte Carlo tree search (MCTS) guided by neural network predictions—not just for policy and value, but also for predicting human actions and the hidden reward parameters.

**Theoretical Claims:**

There are no new theoretical results or novel proofs here. The authors rely on existing results from prior works (e.g., Hadfield-Menell et al. 2016) and cite them appropriately.

---

> ### Author Rebuttal · Authors · 2025-04-01
>
> We thank the reviewer for their thorough review and are glad they found the paper proposes a “novel approach” and that our MBAG environment is “also a strong contribution.” We have responded to individual points and questions below.
>
> ## Small user study
>
> **“The user study is fairly small (only 16 participants), which limits the representativeness of the findings.”**
>
> Our user study was in-person (1-1.5 hours per participant) because we couldn’t run it on a crowdsourcing platform due to latency constraints. Our sample size is comparable to other in-person HCI studies: the median number of participants in CHI 2019 mixed-method studies was 18 [3]. Moreover, our participants had different amounts of experience with Minecraft and computer games, making our sample diverse.
>
> Finally, our results are statistically significant based on appropriate statistical tests. Our assistant enables humans to build the goal with fewer place/break actions (one-sided t test $p \< 0.05$).
>
> ## Markovian human model
>
> **“Your model assumes that human behavior is Markovian… How might your approach perform if humans exhibit non-Markovian behavior”**
>
> We actually do use a non-Markovian human model by employing an LSTM to condition on past observations. Recurrence improved the model’s accuracy (Appendix D.1.2), suggesting humans exhibit non-Markovian behavior. Prior assistance game work also uses recurrent human models [1]. We kept the Markov notation for consistency with the original assistance game formalism.
>
> ## RLHF comparison
>
> **“Have you considered evaluating AssistanceZero against a full RLHF system?”**
>
> Comparing with RLHF would be interesting but challenging because RLHF is not easily applicable to the MBAG environment. First, RLHF is usually formulated as a *single-agent* problem [2, 3], so the additional human agent in our setting would make it difficult to apply standard techniques. Furthermore, in LLMs, RLHF is applied to only a single step of interaction between the assistant and the user, i.e., the comparison data used by RLHF uses conversations which only differ in the last assistant message. In MBAG, the equivalent would be to compare single assistant actions taken in response to a given history of human and assistant actions. However, it may be quite difficult to judge assistant actions in isolation; for instance, more than half of assistant actions are usually movement, and it is unclear how to judge the relative usefulness of say, moving left versus up. For these reasons, we decided to only compare to an SFT baseline, especially since SFT alone for LLMs can often achieve performance close to that of RLHF [7].
>
> Future work could compare assistance games with RLHF in domains where it's more feasible, like LLM post-training.
>
> ## LLM post-training
>
> **“In the real world, humans do not have full observation of the state…”**
>
> Our approach could be applied even if the human is learning about their own reward. The assistant would still be rewarded based on the true goal, and we could train a recurrent human policy on data involving human learning. We could follow approaches like [4], which uses RL in a multi-armed bandit setting where the human learns about their reward parameters through noisy observations. Additionally, the assistant could predict the human’s internal state and learning dynamics to help them learn faster [5]. We also note that such “partially observable assistance games” are studied in [1]. We chose to study the simpler case where the human observes their own goal since it is still challenging.
>
> **“the most efficient approach would be to communicate their intentions to the AI at the beginning (via language) so that the AI can have full observation of the goal.”**
>
> Often, the human cannot fully communicate their goal to the AI. For example, for complex coding tasks, it would be difficult and time consuming to write long, detailed specs. Additionally, if the human wants the code to produce visual output (e.g., a diagram), natural language is insufficient to provide the AI full goal observability. Thus, an assistance game approach would remain beneficial.
>
> Even with perfect communication, an assistant trained with the assistant game framework would likely be more aligned with the human’s true intentions than RLHF, which could deceive humans since it’s trained to seek positive feedback from users [6].
>
> [1] Emmons et al. Observation Interference in Partially Observable Assistance Games
> [2] Christiano et al. Deep reinforcement learning from human preferences
> [3] Reinhard. Participants, Incentives, and User Studies: A Survey of CHI 2019
> [4] Chan et al. The assistive multi-armed bandit
> [5] Tian et al. Towards modeling and influencing the dynamics of human learning
> [6] Williams et al. On targeted manipulation and deception when optimizing LLMs for user feedback
> [7] Zhou et al. LIMA: Less Is More for Alignment.

---

> > ### Comment · Reviewer_YoBP · 2025-04-04
> >
> > Thank you for the clarifications, which have addressed much of my initial concern. However, I am maintaining my current score for two reasons: 1) The size of the human study is still not convincing. 2) If the paper argues AssistanceZero as an alternative to RLHF, it should include a direct comparison. Otherwise, it should refrain from making such a claim.

---

> > > ### Author Response · Authors · 2025-04-08
> > >
> > > We thank the reviewer for their response and we are glad we were able to address some of their concerns.
> > >
> > > Just to summarize our thoughts regarding these two points:
> > > * Our in-person human study is of similar size to comparable work. It’s large enough to show a statistically significant improvement of AssistanceZero over SFT.
> > > * Current RLHF techniques are not directly applicable to our Minecraft domain because it is fundamentally interactive at a granular level in a way that LLM-based chatbots are not. This is why we chose to compare AssistanceZero to SFT. For LLMs, SFT can produce similar results to full RLHF [1], so we think this is a quite strong baseline.
> > >
> > > Thank you again for your time and valuable feedback!
> > >
> > > [1] Zhou et al. LIMA: Less Is More for Alignment.

---

### Decision · Program_Chairs · 2025-05-01

**Decision:**

Accept (poster)

**Comment:**

This paper proposes the use of assistance games as a framework for modern AI agents to infer user goals. The authors propose a benchmark based on Minecraft, and an algorithm (AssistanceZero) based on AlphaZero, that outperforms standard baselines. This offers the opportunity for researchers to leverage the latest improvements in foundation models to train agents that can help humans in unspecified problems, which is an interesting application area and one deserving of more research.